## METHOD

# metaMIC: reference-free misassembly identification and correction of de novo metagenomic assemblies

Senying Lai[1], Shaojun Pan[1], Chuqing Sun[2], Luis Pedro Coelho[1,3*], Wei-Hua Chen[2,4*] and Xing-Ming Zhao[1,3,5,6,7,8*]

*Correspondence:
luis@luispedro.org;
weihuachen@hust.edu.cn;
xmzhao@fudan.edu.cn

[2] Key Laboratory of Molecular Biophysics of the Ministry of Education, Hubei Key Laboratory of Bioinformatics and Molecular-imaging, Center for Artificial Intelligence Biology, Department of Bioinformatics and Systems Biology, College of Life Science and Technology, Huazhong University of Science and Technology, Wuhan, Hubei, China
[3] MOE Key Laboratory of Computational Neuroscience and Brain-Inspired Intelligence, and MOE Frontiers Center for Brain Science, Fudan University, Shanghai, China
[8] Zhangjiang Fudan International Innovation Center, Shanghai, China
Full list of author information is available at the end of the article

## Abstract

Evaluating the quality of metagenomic assemblies is important for constructing reliable metagenome-assembled genomes and downstream analyses. Here, we present metaMIC (https://github.com/ZhaoXM-Lab/metaMIC), a machine learning-based tool for identifying and correcting misassemblies in metagenomic assemblies. Benchmarking results on both simulated and real datasets demonstrate that metaMIC outperforms existing tools when identifying misassembled contigs. Furthermore, metaMIC is able to localize the misassembly breakpoints, and the correction of misassemblies by splitting at misassembly breakpoints can improve downstream scaffolding and binning results.

**Keywords:** Metagenomic assemblies, Misassembled contigs, Misassembly breakpoints, Metagenome-assembled genomes, Binning

## Background

Constructing reliable metagenome-assembled genomes (MAGs) is of great importance for understanding microbial communities and downstream functional analysis, such as taxonomic annotations and reconstruction of metabolic processes [1–4]. MAGs are obtained by binning assembled contigs into bins, the quality of which can be significantly affected by the assembly errors in contigs. For example, the chimerical assemblies consisting of two or more genomes can introduce contamination for reconstructed MAGs, potentially resulting in misleading biological conclusions [5]. Despite the progress in assembly algorithms, errors are still prevalent in metagenomic-assembled contigs owing to the inherent complexity of metagenomic data. Assembly errors including inter- and intra-genome misassemblies are caused by repetitive genomic regions that occur within the same genome or conserved sequences shared among distinct organisms, which is especially likely to happen when multiple closely related strains are present in the same

environment [6, 7]. Therefore, the evaluation of metagenomic assemblies is critical for constructing high-quality and reliable MAGs.

Approaches that have been proposed for assessing the quality of metagenomic assemblies can be grouped into two categories: *reference-based* and *reference-free* approaches. Reference-based methods evaluate the de novo assemblies by aligning them against corresponding reference genomes. For example, MetaQUAST [8], the metagenomic-adapted version of QUAST [9], detects misassemblies such as translocation, inversion and relocation by mapping the metagenomic contigs to a set of closely related reference genomes. However, it is difficult to distinguish errors from real structural variation. Moreover, reference genomes are available for only a small fraction of organisms found in real environments, which limits these approaches to previously sequenced species [10]. Therefore, the evaluation of metagenomic assemblies would benefit from reference-free methods. Typically, these methods exploit features such as the high variation in coverage depth or inconsistent insert distance of paired-end reads to indicate possible repeat collapse, misjoins, or error insertions/deletions [11]. Popular reference-free methods include ALE [12], DeepMAsED [13], SuRankCo [14], and VALET [15]. ALE measures the quality of assemblies as the likelihood that the observed reads are generated from a given assembly by modeling the sequencing process. SuRankCo uses a machine learning approach to provide quality scores for contigs based on characteristics of contigs such as length and coverage. VALET detects misassemblies based on the combination of multiple metrics extracted from the alignment of reads to contigs. Deep-MAsED employs a deep learning approach to identify misassembled contigs. Despite the great value of those approaches for evaluating metagenomic assembly quality, only VALET and ALE predict the position where the misassembly errors are introduced and none of these methods have functionality for correcting metagenomic misassemblies. More importantly, SuRankCo are no longer maintained, and software incompatibilities hinder their use.

Here, we present a novel tool called metaMIC which performs reference-free misassembly identification and correction in de novo metagenomic assemblies. metaMIC can identify misassembled contigs, localize misassembly breakpoints within misassembled contigs, and then correct misassemblies by splitting misassembled contigs at breakpoints. Benchmarking results on both simulated and real metagenomics data show that metaMIC can identify misassembled contigs with higher accuracy than state-of-the-art tools, and precisely localize the misassembly error regions and recognize breakpoints in both single genomic and metagenomic assemblies. By comparing the scaffolding and binning results before and after metaMIC correction, we demonstrate that the correction of misassemblies by metaMIC can improve the scaffolding and binning results.

## Results

### Overview of the metaMIC pipeline

metaMIC is a fully automated tool for identifying and correcting misassemblies in metagenomic contigs using the following three steps (Fig. 1). First, various types of features were extracted from the alignment between paired-end sequencing reads and each contig, including sequencing coverage, nucleotide variants, read pair consistency, and *k*-mer abundance differences (KAD) [16] between mapped reads and the contig. The KAD

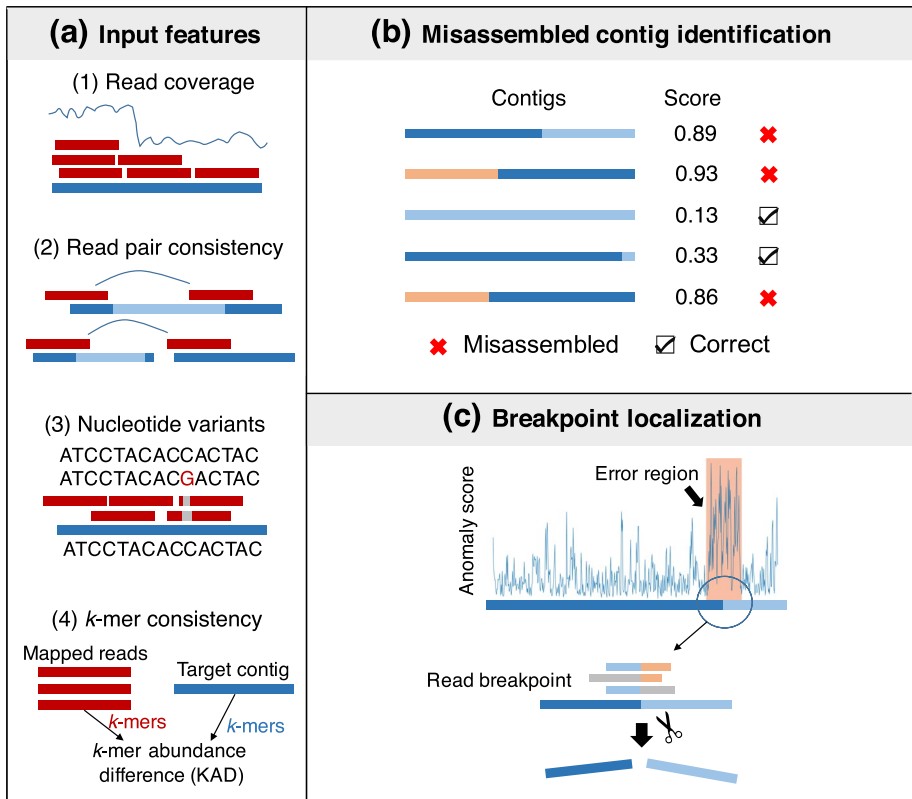

**Fig. 1** The schematic overview of metaMIC. **a** metaMIC extracts four types of features from the alignment of paired-end reads to contigs: read coverage, nucleotide variants, read pair consistency, and *k*-mer abundance consistency. **b** Misassembled contigs are identified by metaMIC based on the four features. **c** metaMIC first identifies the error regions containing misassembly breakpoints, and then recognizes the exact positions of breakpoints and corrects misassemblies by splitting misassembled contigs at breakpoints

method was previously developed for evaluating the accuracy of nucleotide base quality in single genomic assemblies. Here, we extended KAD to metagenomic assemblies to measure the overall consistency between mapped reads and corresponding contigs (see "Methods"). Secondly, the features extracted in the first step are used as input to a random forest classifier for identifying misassembled contigs, where the classifier is trained with simulated bacterial metagenomic communities to discriminate misassembled contigs from correctly assembled ones. Thirdly, metaMIC will localize misassembly breakpoint(s) in each misassembled contig, namely the point at which the left and right flanking sequences are predicted to have originated from different genomes or locations. As most misassemblies are chimeras where two fragments from different locations or with different orientations are mistakenly connected and not just random sequences being generated [9], misassemblies can be corrected by breaking up the contigs into two (or more) correctly assembled contigs.

### Identifying misassembled contigs in simulated metagenomic datasets

To evaluate metaMIC, we tested it on simulated metagenomic datasets obtained from CAMI (the Critical Assessment of Metagenome Interpretation) [2] that comprise a known mixture of organisms. As MEGAHIT [17] can assemble metagenomics data in a

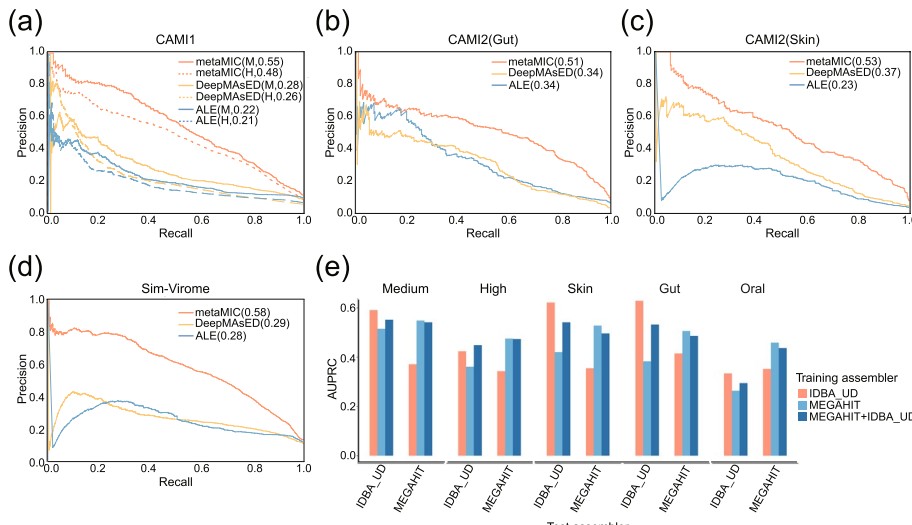

**Fig. 2** metaMIC outperforms ALE and DeepMAsED in identifying misassembled contigs in simulated metagenomic datasets. **a–d** The performance of the three tools on the CAMI-medium (M) and high-complexity (H) communities (**a**), *CAMI2-Skin* (**b**), *CAMI2-Gut* (**c**), and simulated virome dataset (*Sim-Virome*) (**d**). **e** The AUPRC scores of metaMIC on test datasets assembled by MEGAHIT or IDBA_UD (Test assembler), where metaMIC were trained on contigs from training datasets assembled by MEGAHIT, IDBA_UD, or jointly by MEGAHIT and IDBA_UD (MEGAHIT+IDBA_UD)

time- and cost-efficient manner and perform well for metagenomic assembly, we assembled the reads from the CAMI datasets into contigs with MEGAHIT. Since the microbiota composition of those samples is known, we were able to compare the contigs to the reference genomes to obtain the ground truth misassemblies through MetaQUAST. We first evaluated metaMIC on the Medium (*CAMI1-Medium*) and High-diversity communities (*CAMI1-High*) to see how dataset complexity will influence the accuracy of metaMIC. We noticed that the types of misassemblies identified in these two datasets were slightly different, and the CAMI1-High dataset contains more inter-genome translocations and higher proportion of misassemblies while the CAMI1-Medium dataset contains more relocations (see Additional file 1: Fig. S1-2), which is consistent with previous conclusion that datasets with higher intra-species diversity tend to have more inter-genome translocation misassemblies [13]. Compared with CAMI1-High metaMIC performed better on (Fig. 2a; although still significantly better than existing tools) CAMI1-Medium, implying that the higher microbial diversity increases the challenge of identifying misassembled contigs. We further compared metaMIC on these datasets against ALE [12] and DeepMAsED [13] (see "Methods"). As shown in Fig. 2a, metaMIC significantly outperforms both ALE and DeepMAsED on the two datasets, as metaMIC achieved the highest AUPRC (area under the precision-recall curve).

We also evaluated metaMIC and other tools on simulated metagenomic datasets from three different human body sites: gastrointestinal tract (*CAMI2-Gut*), skin (*CAMI2-Skin*), and oral cavity (*CAMI2-Oral*). As shown in Fig. 2b, c and Fig. S3, metaMIC has the highest precision when identifying misassembled contigs at any recall threshold. The higher inter-genome similarity in CAMI2-Oral may help explain the lower AUPRC achieved by metaMIC (Additional file 1: Fig. S3). When looking at the misassembled contigs identified by metaMIC, we noticed that metaMIC performs better on shorter

contigs and inter-genome translocation misassemblies (Additional file 1: Fig. S4-5). Additionally, we tested metaMIC on a simulated virome datasets (*Sim-Virome*), which were simulated based on 1000 complete viral genomes randomly selected from NCBI RefSeq collection [18] (see "Methods"). The Sim-Virome contains mainly translocations and relocations but few inter-genome translocations and inversions. We found that metaMIC still significantly outperforms both ALE and DeepMAsED on Sim-Virome dataset as shown in Fig. 2d, indicating that metaMIC can also be used for virome assemblies besides bacterial metagenomic assemblies.

   As metaMIC can be trained on contigs assembled by different assemblers, we further investigated the impact of different assemblers on the performance of metaMIC when identifying misassembled contigs. Here, two popular assemblers, i.e., MEGAHIT and IDBA_UD [19], used for metagenomic data were considered. As shown in Fig. 2e, we found that metaMIC performed best when it was trained on the same assembler as it was later evaluated, where MEGAHIT and IDBA_UD means that metaMIC is trained with the union set of contigs assembled by those two tools. The influence of different assemblers may be due to the distribution difference of misassembly types in the contigs generated by the two assemblers. IDBA_UD [19] is shown to generate more intra-genome translocations while MEGAHIT is shown to generate more inter-genome translocations (Additional file 1: Fig. S6). Therefore, we recommend to use metaMIC trained on the contigs generated by the same assembler. Right now, metaMIC provides built-in models for MEGAHIT, IDBA_UD, and metaSPAdes [20] and can be used to generate new models based on the assembler specified by users (see Additional file 1: Fig. S7 for the metaSPAdes specific model tested on CAMI datasets).

### metaMIC can identify breakpoints with higher accuracy in misassembled contigs

Beyond identifying misassembled contigs, metaMIC is able to accurately recognize the misassembly breakpoints, at which the misassembled contigs can be split into shorter ones. In the CAMI datasets, the misassembly breakpoints mostly occur in the coding sequences, which accounted for approximately 65% of all breakpoints (see Additional file 1: Fig. S8), indicating that most misassemblies may disrupt gene structures and in turn influence the downstream functional analysis. To localize the misassembly breakpoints, metaMIC will first scan each contig with a sliding window of 100bp and calculate an anomaly score for each 100bp region by employing isolation forest [21], where the region with a higher anomaly score is possibly an error region containing a misassembly breakpoint. In this way, metaMIC is able to localize the misassembly breakpoint in a specific short region (Fig. 3a). From the distribution, we can clearly see that the error regions containing any misassembly type generally have significantly higher anomaly scores than error-free regions, and the inter-genome translocation error is most prevalent in the dataset. It is indeed the inter-genome translocation error that occurs most often in the CAMI datasets (Additional file 1: Fig. S2). The differential distribution of anomaly scores between error and error-free regions implies that the anomaly score has the potential to recognize the error regions. We also noticed that the misassembly sites are usually read breakpoints (locations at which the boundaries of aligned read fragments do not coincide with the ends of corresponding reads) [22]. Similar to anomaly scores, we found that the read breakpoint ratio was significantly different between error

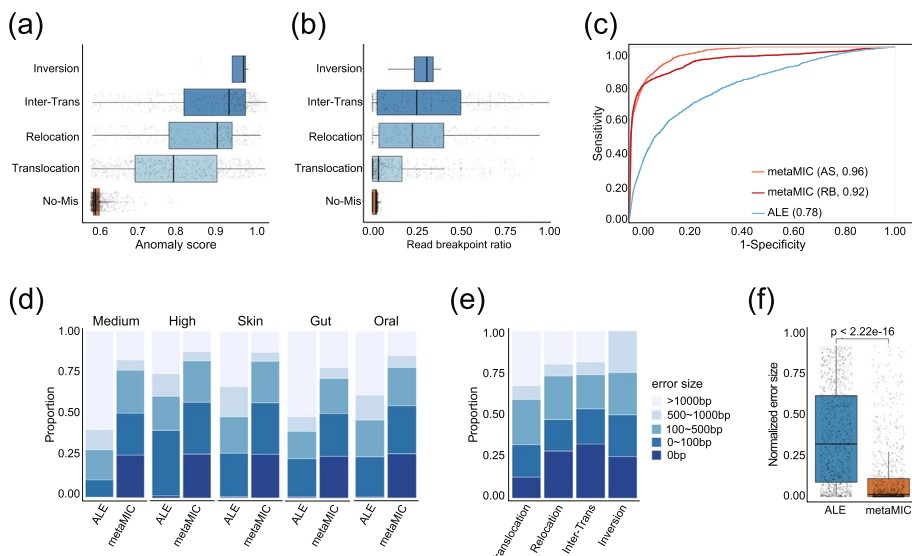

**Fig. 3** The performance of metaMIC in localizing misassembly breakpoints on CAMI datasets. **a, b** The distribution of anomaly scores (**a**) and read breakpoint ratios (**b**) of different misassembly types across contigs from CAMI1-Medium. **c** The receiver operation curves by ALE, anomaly scores (AS), and read breakpoint ratios (RB) when discriminating error regions from error-free regions in CAMI1-Medium, respectively. **d, e** The distribution of error size of misassembly breakpoints recognized by metaMIC on CAMI1-Medium (*Medium*), CAMI1-High (*High*), CAMI2-Skin (*Skin*), CAMI2-Gut (*Gut*), and CAMI2-Oral (*Oral*) (**d**), and different misassembly types in CAMI1-Medium (**e**). **f** The distribution of normalized error size of misassembly breakpoints recognized by metaMIC and ALE on CAMI1-Medium

regions and error-free regions (Fig. 3b, see also Additional file 1: Figs. S9-10), where the read breakpoint ratio is defined as the ratio of fragmentally aligned reads to the total number of mapped reads.

Due to the potential of read breakpoint ratio and anomaly score to localize the error regions, we want to see whether metaMIC can use these two features to separate the erroneous regions from error-free regions. From the receiver operation curves shown in Fig. 3c, we can see that with either anomaly score or read breakpoint ratio, metaMIC can classify the error regions containing misassembly breakpoints with error-free regions more accurately than ALE. To combine the usages of these two features, metaMIC first localizes the error regions in a misassembled contigs with the help of anomaly score, and then identifies the exact breakpoints in an error region based on the read breakpoint ratio.

We evaluated the performance of both metaMIC and ALE on the five datasets from CAMI as shown in Fig. 3d. We observed that approximately 71–86% of the metaMIC-predicted breakpoints were within 500bp compared to 26–48% of those by ALE. More importantly, metaMIC could predict the exact locations for ~25% of the breakpoints with the use of read breakpoints. Again, inter-genome translocations or inversions can be detected with higher accuracy relative to other misassembly types (Fig. 3e), consistent with previous results that they were supported by more fragmentally aligned reads and had higher anomaly scores as compared with other error types (see Fig. 3a, b; Additional file 1: Fig. S10). Given the possible influence of contig length on the prediction error size, we normalized the error size by the contig length and compared the results of metaMIC

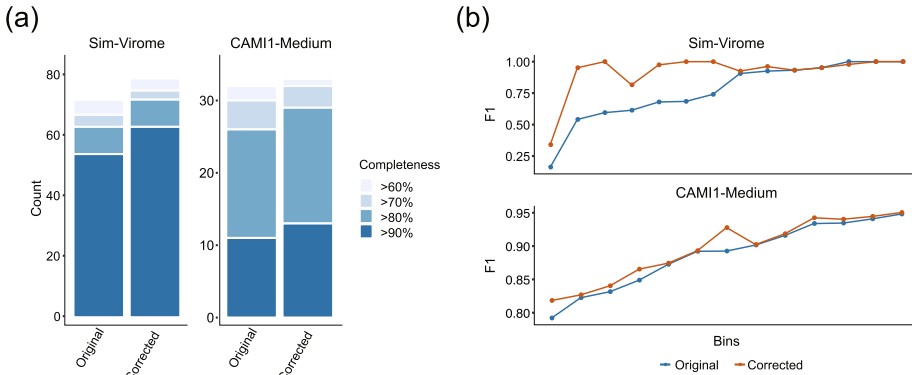

**Fig. 4** Splitting misassembled contigs at breakpoints improves the downstream binning results over *Sim-Virome* and *CAMI1-Medium* datasets. **a** The number of high-quality bins with low contamination (<5%) of different completeness reconstructed from original and corrected contigs. **b** The distribution of *F*1 scores for bins reconstructed based on contigs before and after correction, where only those bins whose results change before and after correction were shown for clearness

with those of ALE. As shown in Fig. 3f, metaMIC still significantly outperforms ALE with respect to the normalized error size (Wilcoxon's test *P*-valu*e* <2.22e−16), where the median and mean of the metaMIC's normalized error size were 0.01 and 0.11, respectively, compared to 0.39 and 0.34 for ALE (see also Additional file 1: Fig. S11). The similar results can also be obtained on contigs assembled by IDBA_UD and metaSPAdes (see Additional file 1: Figs. S12-16).

**Splitting misassembled contigs improves downstream binning performance**

As metaMIC can identify breakpoints in misassembled contigs, it can split misassembled contigs at breakpoints and reduce the number of misassemblies (see "Methods"), although the contiguity could be slightly decreased due to more fragmented contigs [23]. To see how the correction of splitting misassembled contigs at breakpoints employed by metaMIC will influence downstream analyses, we binned the contigs in the simulated datasets using MetaBAT2 [24]. The misassembled contigs were first identified by metaMIC from these datasets. After splitting the misassembled contigs at breakpoints, the new contigs were treated as corrected contigs. We then assessed the binning performance over the original and metaMIC-corrected contigs by counting the number of obtained high-quality bins. We can see in Fig. 4a that metaMIC correction increases the number of near-complete reconstructed bins (completeness above 90%, contamination below 5% [3]) by 10–20% (see also Additional file 2: Table S1), showing that the correction of metagenomic miassemblies has significant impact on downstream binning. We noticed that most of the misassemblies corrected by metaMIC were inter-genome translocations that were also the main sources of chimeras and assembly errors in CAMI datasets (Additional file 1: Fig. S2; Additional file 2: Table S2). From Fig. 4b, we can see that the bin-wise *F*1 scores of those bins constructed from corrected contigs are significantly improved compared with the results over original contigs (paired Wilcoxon's test *P*-value <0.05, see Additional file 2: Table S1), indicating that the reconstructed bins can better represent the reference genomes after metaMIC correction. The above results clearly demonstrate that the correction of metagenomic misassemblies by metaMIC can

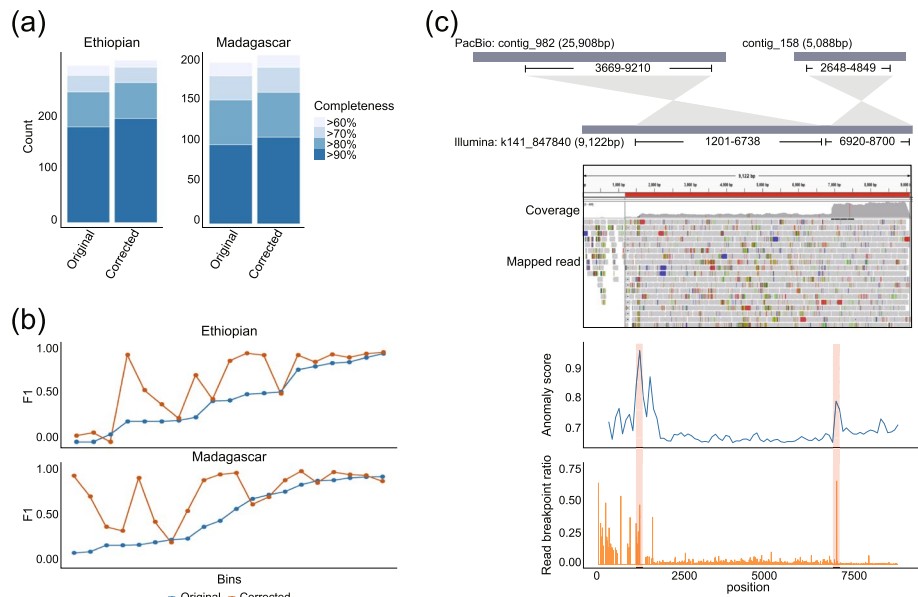

**Fig. 5** The performance of metaMIC on real metagenomic datasets. **a** The number of bins of different completeness with low contamination (<5%) reconstructed from original and corrected assemblies of "*Ethiopian*" (left) and "*Madagascar*" (right) cohorts. **b** Comparison of *F*1 scores for reconstructed bins before and after correction of contigs from "*Ethiopian*" (top) and "*Madagascar*" (bottom) cohorts. **c** An example of a predicted misassembled contig "k141_847840" assembled from combined rumen fluid and solid sample. The top plot shows the alignment result of Illumina short-read assembled contig "k141_847840" and PacBio long-read assembled contigs ("contig_982" and "contig_158"), where two regions in the "k141_84780" (1201-6738bp and 6920-8700bp) were aligned to "contig_982" and "contig_158," respectively. The middle figure shows a snapshot of Integrative Genomics Viewer for contig "k141_847840." The bottom plot shows the anomaly score (blue) and read breakpoint ratio (orange) across contig "k141_847840"

significantly improve the resulting bins in term of both completeness and contaminations, which is important for understanding the complex microbiota communities.

### Application of metaMIC to real metagenomic datasets

To better evaluate the performance of metaMIC, we applied metaMIC to two recent human gut metagenomics datasets from Ethiopian [25] and Madagascar [26] cohorts that consist of 50 and 112 samples, respectively. In total, metaMIC respectively identified 5905 and 18,436 misassembled contigs in *Ethiopian* and *Madagascar* datasets, which represents 2.59 and 4.53% of all contigs in the two datasets. We then separately binned the original and corrected contigs into bins using MetaBAT2. Strikingly, we found that ~20% of the resulting original bins contained misassemblies, although the latter accounted for less than 5% of all contigs. As previous results have demonstrated that metaMIC correction can improve the binning results in simulated datasets, we further explored whether the correction step employed by metaMIC can improve downstream results in real datasets. As shown in Fig. 5, in addition to obtaining more bins of high-quality (Completeness >90 and Contamination <5) (Fig. 5a), most of the corrected bins had an equal or higher *F*1 scores compared to the corresponding original bins (Fig. 5b, paired Wilcoxon's test *P*-value <0.05, see Additional file 2: Table S3). The results indicate that the misassembled contigs identified by metaMIC in these

two real datasets are really misassembled, the correction of which can significantly improve downstream analysis results.

As these contamination metrics are based on in silico evaluation, we further tested the ability of identifying misassemblies using another metagenomic dataset (a combined rumen fluid and solid sample, SRA BioProject numbers *PRJNA507739*) where both short and long reads are available. Since the long reads from PacBio platforms are able to span repeats [27, 28], which are the main contributor to misassemblies, we can therefore use the long-read assemblies as gold standards to validate our predicted misassembled contigs from the short-read assemblies. Approximately 10% of the short-read assemblies can be aligned to long-read assemblies, among which a total of 93 misassembled contigs were detected. When testing on those misassembled contigs, metaMIC is able to correctly identify 43 misassemblies without false negatives. As an example, there exist two peaks at positions of 1200 and 6920bp in the contig of "k141_847840" according to the anomaly scores by metaMIC, and both peaks, especially the one at 6920bp, contain higher read breakpoint counts implying possible misassembly breakpoints at these two locations. When aligning this contig against the long-read assemblies, we found that two regions in this contig (1201–6738bp and 6920–8700bp) were indeed aligned to two different long-read assembled contigs, and a change-point in the read coverage at 6920bp can be observed (see Fig. 5c), indicating that there are actually two contigs wrongly assembled into one contig at position of 6920bp. We also found that only a few reads can be aligned to the region of 0–1200bp, suggesting this region may be extended mistakenly by the assembler. Thus, the misassemblies predicted by metaMIC in the contig of "k141_847840" were indeed assembly errors. Furthermore, although some identified misassemblies by metaMIC could not be validated by long-read assemblies, the visualization results of paired-end reads mapping to contigs were able to demonstrate the predictions (see Additional file 1: Fig. S17). We also evaluated metaMIC on a mock microbiome community (GIS20, SRA BioProject numbers *PRJEB29139*) which were sequenced from a mixture of 20 cultured strains of bacteria [29]. As shown in Fig. S18, metaMIC significantly outperforms existing tools in identifying misassembled contigs and localizing misassembly breakpoints. Approximately 65% misassembly breakpoints can be localized with predicted error size smaller than 500bp (Additional file 1: Fig. S18). In summary, all these results shown above demonstrated the effectiveness of metaMIC when applied to real metagenomic datasets.

### Application of metaMIC to isolate genomes

Since metaMIC can identify and correct intra-genome misassemblies such as inversions and relocations, metaMIC can also be applied to isolate genomes. We tested metaMIC on four real datasets from GAGE-B project [30], which aimed to evaluate assembly algorithms on isolate genomes. We tested metaMIC on *B. cereus*, *M. abscessus*, *R. sphaeroides*, and *V. cholerae*, where the raw reads, assembled contigs [31], and curated reference genomes are available for these four species. metaMIC was ran on the assemblies downloaded from GAGE-B project and its performance was evaluated with the results by QUAST [9] as gold standard. These four datasets contain mainly relocations but a few translocations. Slightly different from metagenomes, the location of misassembly breakpoints will be detected directly on all contigs instead of only

misassembled contigs, thus misassembled contig identification step and the training process is not needed for metaMIC when applied to isolate genomes. In more details, metaMIC first calculates an anomaly score for each window of 100bp to localize the error region on all contigs, and the regions with anomaly scores >0.95 (see "Methods") are likely error regions. Then the breakpoints will be recognized in the error regions by using read breakpoint ratios (see "Methods"). We noticed that similar to metagenomes, the error regions in isolate genomes also have higher anomaly scores and more read breakpoints than error-free regions (Additional file 1: Fig. S19-20). We then compared metaMIC against MEC [32], a recently developed misassembly correction tool, when identifying misassembly breakpoints on the four isolate genomes. As shown in Table 1, metaMIC identified more true misassemblies than MEC, where approximately 80% misassemblies can be corrected compared to ~30% of MEC, and after the correction by metaMIC, the total number of bases of uncorrected misassembled contigs (i.e. misassembled contig length in Table 1) was significantly reduced compared with that by MEC.

To further see influence of misassembly correction on isolate genomes, we scaffolded original and corrected contigs separately with popular scaffolders including BESST [33] and ScaffMatch [34], and then used QUAST to evaluate the scaffolding results. As seen in Table 2 and Table S4, the number of misassemblies in the scaffolding results based on metaMIC's corrected contigs was much lower than that based on the original uncorrected contigs, and metaMIC significantly outperforms MEC in terms of misassembled contig length. Moreover, metaMIC performs comparably well or better compared against MEC in terms of NA50 and total aligned length and performs better especially for *R.sphaeroides*. The above results clearly show the effectiveness of metaMIC when identifying and correcting misassembled contigs on isolate

**Table 1** Performance comparison of metaMIC and MEC on four real datasets from the GAGE-B project

| Species | Correction tool | Misassembled contig length | FN | TP | FP |
|---|---|---|---|---|---|
| *M. abscessus* | raw | 1,189,973 | 20 | \ | \ |
| | MEC | 982,986 | 14 | 6 | 2 |
| | metaMIC | 593,741 | 6 | 14 | 2 |
| *V. cholerae* | raw | 597,777 | 7 | \ | \ |
| | MEC | 597,183 | 6 | 1 | 0 |
| | metaMIC | 205,644 | 3 | 4 | 1 |
| *R. sphaeroides* | raw | 135,153 | 2 | \ | \ |
| | MEC | 64,489 | 1 | 1 | 0 |
| | metaMIC | 0 | 0 | 2 | 7 |
| *B. cereus* | raw | 117,830 | 5 | \ | \ |
| | MEC | 56,086 | 4 | 1 | 0 |
| | metaMIC | 28,068 | 1 | 4 | 3 |

Misassembled contig length denotes the total number of bases in the raw misassembled contigs or the misassembled contigs that cannot be corrected by MEC or metaMIC. True positive (TP) is the number of true misassemblies identified by the error correction tool. False positive (FP) is the number of misassemblies which are actually correct but mistakenly identified as misassemblies. False negative (FN) denotes the number of true misassemblies that are not identified

**Table 2** Comparison of BESST scaffolding results of contigs before and after correction

| Species | Correction tool | #Contigs | #Total aligned length | Total length($\geq$0bp) | Total length($\geq$1000bp) | Misassembled contig length | NA50 | #Mis |
|---|---|---|---|---|---|---|---|---|
| M. abscessus | raw | 262 | 5,045,398 | 5,160,404 | 5,129,190 | 1,303,084 | 45,957 | 27 |
| | MEC | 268 | 5,045,445 | 5,160,476 | 5,129,015 | 982,986 | 40,129 | 24 |
| | metaMIC | 274 | 5,045,398 | 5,160,404 | 5,129,190 | 755,186 | 47,488 | 17 |
| V. cholerae | raw | 201 | 3,936,390 | 3,958,533 | 3,921,645 | 597,777 | 43,122 | 10 |
| | MEC | 202 | 3,935,796 | 3,958,533 | 3,921,645 | 597,183 | 43,122 | 9 |
| | metaMIC | 205 | 3,936,390 | 3,958,533 | 3,921,645 | 205,644 | 43,123 | 7 |
| R. sphaeroides | raw | 231 | 4,492,687 | 4,519,491 | 4,486,060 | 359,217 | 75,728 | 5 |
| | MEC | 230 | 4,492,749 | 4,519,550 | 4,486,119 | 168,646 | 78,611 | 4 |
| | metaMIC | 232 | 4,493,107 | 4,520,061 | 4,486,630 | 51,809 | 78,920 | 3 |
| B. cereus | raw | 141 | 5,310,597 | 5,381,347 | 5,369,165 | 332,560 | 104,970 | 7 |
| | MEC | 140 | 5,310,816 | 5,381,940 | 5,369,758 | 332,001 | 104,970 | 7 |
| | metaMIC | 140 | 5,311,395 | 5,382,650 | 5,370,789 | 175,743 | 104,970 | 5 |

#Mis denotes the number of scaffolds that contain misassemblies. Total aligned length denotes the length of total number of bases from contigs that can be aligned to the assembly

genomes, and also the capability of maintaining or improving the contiguity of downstream scaffolding after correction.

## Discussion

We present a novel tool named metaMIC to identify and correct misassembled contigs from de novo metagenomic assemblies and demonstrate its effectiveness on both simulated and real metagenomic datasets of varying complexity. Unlike most existing metagenomic assembly evaluation methods that only evaluate individual contigs or overall assemblies, metaMIC is capable of localizing the misassembly breakpoints and then corrects the misassembled contigs at breakpoints. By integrating various types of features extracted from both reads and assemblies, including read coverage, read pair consistency, nucleotide variants, and *k*-mer abundance consistency, metaMIC is able to detect intra- and inter-genome misassemblies. Additionally, metaMIC can also be applied on isolate genomes given its ability in identifying intra-genome misassemblies. After the correction of misassemblies, metaMIC can significantly help improve the performance of downstream analysis including binning and scaffolding. Although metaMIC was trained on the bacterial metagenomes in this work, metaMIC can be easily transferred to archaea, eukaryotes, or viruses since it only relies on the features extracted from the mapping results between paired-end reads and assembled contigs. For example, metaMIC model trained on the bacterial metagenomes can work well on simulated virome datasets (Fig. 2d). In the future work, we will consider training metaMIC specifically for archaea, eukaryotes, or viruses.

In this study, the performance of metaMIC is mainly shown on the metagenomic assemblies assembled by MEGAHIT due to its high memory efficiency [35]. metaMIC can also be applied to metagenomic assemblies constructed by IDBA_UD and metaSPAdes (see Fig. 2e and Additional file 1: Fig. S7). As different assemblers tend to be biased to certain types of misassemblies, the models trained on the outputs of one assembler may not transfer well to other assemblers. Note that metaMIC can be easily

extended to work on the metagenomic assemblies by other assembler tools if the training datasets generated by the corresponding assemblers are available. We suggest to use metaMIC on the datasets from the same assembler as the one it is trained on.

metaMIC scans each contig with a sliding window of 100bp to localize the candidate error regions. Generally, the accuracy of localizing misassembly breakpoints decreases as the sliding window size increases, thus a shorter window size has a higher resolution to pinpoint error regions (Additional file 1: Fig. S21). However, more computation resources will be required if the sliding window becomes shorter. In addition, metaMIC currently only works on misassembled contigs containing a single misassembly since multi-error containing contigs only make up a small proportion of all misassembled contigs (see Fig. S22). In the future, the multi-error containing contigs will also be taken into account by metaMIC.

metaMIC correction mainly relies on splitting contigs at misassembly breakpoints. However, caution should be needed here as more fragmented sequences will be generated and mistakenly splitting may result in disrupted gene structure, which can have adverse influence on downstream functional genomic analysis. This phenomenon especially will happen in multi-error containing contigs. Although we have showed that metaMIC correction can improve the downstream binning results, the quality of reconstructed draft bins can be further improved if the broken contigs can be joined into scaffolds correctly. Thus, the combination of metaMIC and scaffolding algorithms will be a promising direction for future research, leading to effective approaches for reconstructing genomes from sequencing data with higher quality and completeness.

Several directions hold promise for further improvements to metaMIC. Firstly, metagenomic read mapping can be evaluated in more robust manner by aligning multi-assigned reads in a probabilistic manner to their contig of origin [36] or using base-level quality metrics such as CIGAR strings [37]. Secondly, the repetitive characteristics in contigs can be used as a candidate misassembly signature in the future. We noticed that a significantly higher proportion of misassembled contigs contained short tandem repeats than the correctly assembled ones, and more than 10% of misassembly breakpoints occurred in the repeat regions (Additional file 1: Fig. S8). Thirdly, the factors that may result in false positive predictions, such as structural variation within species of high similarity and G-C bias in sequencing coverage could be taken into account in future work. Finally, as reference genomes of many bacteria are available, a better performance can be achieved by the combination of reference-free and reference-based approaches.

## Conclusions

Here, a novel tool named metaMIC is developed for identifying and correcting misassemblies in de novo metagenomic assemblies without the use of reference genomes. Benchmarking on both simulated and real datasets, we show that metaMIC is able to pinpoint misassemblies in both single and metagenomic assemblies. We also demonstrate that metaMIC is able to improve the scaffolding or binning results by splitting misassembled contigs at misassembly breakpoints. As none of current assemblers can achieve a completely accurate assembly and misassemblies in contigs have negative

influence on downstream analysis, we expect metaMIC can serve as a guide in optimizing metagenomic assemblies and help researchers be aware of problematic regions in assembled contigs, so as to avoid misleading downstream biological analysis.

## Methods

### metaMIC workflow

metaMIC is implemented in Python3 (Python = 3.7) and runs under Linux with modest running time and memory usage (Additional file 2: Table S5). It requires assembled contigs in FASTA format and paired-end reads in FASTA or FASTQ format as input. Alternatively, the user can provide a BAM file with read pairs mapping to contigs. Given the contigs, metaMIC will first identify the misassemblies by employing a random forest classifier trained on the features extracted from reads and contigs. Next, metaMIC will identify the regions containing misassembly breakpoints in the misassembled contigs based on the anomaly scores, and then recognize the exact positions of the breakpoints in the error regions. Then metaMIC will correct the misassemblies by splitting the contigs at the breakpoints. The details will be given below.

### *Features extracted from reads and contigs*

BWA-MEM (v.0.7.17) [38] is used to map paired-end reads to assemblies, followed by using samtools (v1.9) [39] to filter low-quality mappings and sorting the alignments (see more details in Additional file 1). Then the features will be extracted from the sorted BAM file and can be categorized into four feature types, including read pair consistency, read coverage, nucleotide variants, and *k*-mer abundance consistency. The features belonging to each feature type are explained in detail below (see full description of features in Additional file 1).

For each paired-end reads with left and right mate reads, the insert size corresponding to the distance between two mates is assumed to follow a normal distribution $N(\mu, \sigma)$ [32]. The expected insert size ($\mu$) was calculated as the median value of all insert sizes of read pairs, whereas the standard deviation ($\sigma$) was estimated by the median absolute deviation of insert sizes (Additional file 1). A read is regarded as a proper read if the insert size belongs to $[\mu - 3\sigma, \mu + 3\sigma]$ and the orientation is consistent with its mate, and is a discordant read otherwise. Discordant reads can be further divided into three types: reads with their mates mapped to different contigs, reads with wrong insert size, and reads with orientation not consistent with their mates. A read is regarded as a clipped read if it contains at least 20 unaligned bases at either end of the read, and a read is regarded as a supplementary read if different parts of the read are aligned to different regions of contigs. Given a contig, metaMIC will calculate the proportion of above six types of reads among all reads mapping to the contig as six read features. The same will work for a given region from a contig when identifying error regions.

The coverage-based statistics including read coverage and fragment coverage are calculated at each base of the contig. The read coverage per base represents the number of reads that are mapped over that base, and the fragment coverage is the number of proper paired-end reads spanning that base. Both the read coverage and fragment coverage at each base $i$ of contig $c$ are further standardized as follows.

$$Standardized\_read\_coverage_{ic} = \frac{read\_coverage_{ic}}{\left(\sum_{j=1}^{L_c} read\_coverage_{jc}\right)/L_c},$$

$$Standardized\_fragment\_coverage_{ic} = \frac{fragment\_coverage_{ic}}{\left(\sum_{j=1}^{L_c} fragment\_coverage_{jc}\right)/L_c},$$

where $L_c$ is the length of contig *c*. Then the standardized deviations of coverage and fragment coverage for contig *c* are used as input of metaMIC as shown below:

$$\sigma_{read\_coverage_c} = \sqrt{\frac{\sum_{j=1}^{L_c}\left(standardized\_raad\_coverage_{jc} - 1\right)^2}{L_c}},$$

$$\sigma_{fragment\_coverage_c} = \sqrt{\frac{\sum_{j=1}^{L_c}\left(standardized\_fragment\_coverage_{jc} - 1\right)^2}{L_c}}.$$

metaMIC will calculate the proportion of the bases with unusual coverage as well as the standardized deviations of coverage for each contig, and the same for a given region (see Additional file 1).

The nucleotide variant information is extracted from BAM file with the help of samtools. metaMIC counts the number of discordant, ambiguous, and correct alignments separately at each position. For example, the nucleotide at position *j* in the assembled contig is A and ten reads are mapped to that position, with six having A and three having T, another one having ambiguous base (N) at position *j*. Then for that position, the number of discordant, correct, and ambiguous alignments is three, six, and one, respectively. For each type of alignment in a contig, metaMIC will calculate the proportion of the alignment by dividing the number of this type of alignments to the total number of mapped bases across the contig, and the same for a given region.

metaMIC calculates the *k*-mer abundance difference (KAD) at each base based on the alignment of paired-end reads to contigs. The KAD value, proposed by He et al. [16], measures the consistency between the abundance of a *k*-mer from short reads and the occurrence of the *k*-mer in the genome. Here, for a given *k*-mer and a contig, the copies of the *k*-mer in the contig is *n*, and the *k*-mer abundance in the mapped paired-end reads is *c*, and the sequencing depth of the contig is *m* calculated based on the occurrence of single-copy *k*-mers in the reads. Thus, the KAD value is defined as follows.

$$KAD\ value = \log_2 \frac{(c + m)}{m(n + 1)}$$

A *k*-mer with KAD value not belonging to [−0.5, +0.5] will be regarded as an error *k*-mer, and a base is regarded as an error base if an error *k*-mer covers that base. For a given contig, metaMIC will count the number of error bases across the contig and divide it by the contig length. The proportion of error bases within a given region from a contig will be calculated in the same way.

In summary, the above these four types of features will be extracted for the whole contig (contig-based features) or a window of 100bp (window-based features). The contig-based features will be used to train a random forest to identify misassembled contigs, while the window-based features will be used as input of isolate forests to recognize the error regions containing breakpoints.

### Identification of misassembled contigs

With the above contig-based features, metaMIC trains a random forest [40] with chosen parameters implemented in Scikit-Learn [41] to discriminate misassembled contigs from those correctly assembled ones, where an ensemble of 1000 trees are used. For each contig, a probability score $S(c)$ representing the likelihood that the contig $c$ is misassembled will be output by metaMIC.

### Model training and optimization

We split the data into two parts, and 2/3 of the data was used as the training set (20 metagenomes) for optimizing the parameters and 1/3 of the data used as the test set (10 metagenomes) for evaluating the learned models, where the test set will not be seen or used during the training process. The random forest model was trained on a training dataset, whereas the ground truth misassembly label of contigs provided by metaQUAST is used as a target for training the model. We performed 10-fold cross-validations on the training set to select the optimal hyperparameters for the random forest classifier. In more detail, given 20 metagenomes as the training set, we randomly pick 2 of them for validation while the rest 18 metagenomes for training. This procedure will be repeated for 100 times to make sure the robustness of the model trained. To handle the imbalance problem existing between the positive and negative samples, we will perform down-sampling to obtain the same number of correct contigs as that of the misassembled contigs, where there are fewer misassembled contigs as positive samples. Specifically, a subset of negative samples will be randomly selected from the whole negative training set during the down-sampling procedure and the size of the negative subset is the same as the positive training set. This down-sampling will be repeated for 10 times, and a classifier will be trained based on the positive training set and a selected negative subset each time. Thus, an ensemble classifier will be obtained with the average result of the 10 trained classifiers as output, and the ensemble classifier will be used as the final classifier.

We used the AUPRC, corresponding to the area under the precision-recall curve, to evaluate the performance of the trained classifier, and the parameters leading to the highest AUPRC during the cross-validation procedure will be used as the optimal hyperparameters (See Additional file 2: Table S6). In addition to AUPRC, area under the receiver operating characteristic curve (AUC) and out-of-bag (Oob) error rate were also used to evaluate metaMIC. The results of the 10-fold cross-validation experiments were provided in Table S7, where the 10-fold cross-validation results averaged over 100 times where shown. To evaluate metaMIC's generalization ability, we also assessed its performance on the test set that were not used for training (Additional file 2: Table S7).

### Localizing breakpoints in misassembled contigs

After identifying misassembled contigs, metaMIC is able to localize the misassembly breakpoints in those misassembled contigs. Firstly, metaMIC scans each contig with a sliding window of 100bp and calculates an anomaly score for each window by employing isolation forest [21] based on window-based features to localize the error regions containing misassembly breakpoints, where the region with a higher anomaly score may be an error region. Secondly, metaMIC uses the read breakpoint ratio to recognize the exact misassembly breakpoint in an error region. Specifically, for a given predicted misassembled contig, metaMIC identifies a 100-bp region with the highest anomaly score as an error region and then the position with the highest read breakpoint ratio within this window as the misassembly breakpoint. For those error regions without read breakpoints, the central position of the error region is regarded as the misassembly breakpoint. Note that the 300-bp regions at both ends of a contig are not considered here due to the poor mapping quality.

### Correcting misassemblies by splitting at breakpoints

For a given misassembled contig identified above, metaMIC is able to correct misassembled contigs by splitting the contig into parts at the breakpoint recognized above. To avoid splitting too many contigs mistakenly, only those misassemblies with score of 0.8 (also the default setting for metaMIC) are corrected, leading to the precision of 70~80%. Moreover, metaMIC only corrects contigs that can be split into fragments no shorter than 1000bp in order to avoid generating too short sequences.

### Identifying misassemblies in isolate genomes

When applying metaMIC on isolate genomes, metaMIC detects misassembly breakpoints on all contigs directly without identifying misassembled contigs due to the lower complexity and fewer genomes compared to metagenomics. Specifically, metaMIC first calculates anomaly score for each 100-bp window region across all contigs, and the windows with anomaly scores >0.95 are regarded as error regions, and the positions with read breakpoint count >5 and read breakpoint ratio >0.2 within these error regions are predicted as misassembly breakpoints. The thresholds of anomaly score and read breakpoint ratio were chosen as the ones that lead to the highest $F$1 score (See Additional file [1]: Fig. S24).

### Datasets

The training datasets used for training metaMIC contains 30 bacterial metagenomes which were generated in the same way as DeepMAsED [13]. In more detail, 1000 bacterial genomes with high-quality (CheckM estimated completeness >90 and contamination <5) from the Genome Taxonomy Database [42] were used for generating training datasets as DeepMAsED did (Additional file [1]: Fig. S25). Then MGSIM [13] was utilized to create 30 replicate metagenomes in which $10^7$ Illumina HiSeq2500 150bp paired-end reads were simulated per metagenome with the ART-defined default error distribution [43]. Abundance of each bacterial genome is sampled based on a lognormal distribution *Lognormal*(5, 2). Contigs are assembled with MEGAHIT, IDBA_UD, or metaSPAdes [20]

to generate assembler-specific training datasets, and the contigs <5000bp were removed (see in Additional file 1 for parameter settings of assemblers).

To evaluate the performance of different tools, another six simulated and four real datasets are used to see the performance of metaMIC and other competing tools in identifying misassemblies. The simulated virome dataset were generated in the same way as the method used to generate the training dataset. It contains 10 replicate virome metagenomes simulated based on the 10,140 complete viral genomes obtained from NCBI RefSeq collection [18], where each replicate contains 1000 viral genomes. The other five simulated datasets are obtained from CAMI challenge [2], including the medium-complexity (132 genomes) and high-complexity (596 genomes) communities from CAMI1, CAMI2 Gastrointestinal (130 genomes), CAMI2 Oral (327 genomes), and CAMI2 Skin (233 genomes) from CAMI2. For all datasets, the synthetic short paired-end reads were assembled using MEGAHIT, IDBA_UD, or metaSPAdes. Only contigs >5000bp are considered here as the majority of misassemblies occurred in contigs are longer than 5000bp (Additional file 1: Fig. S26-27). As the ground truth, the misassemblies detected by MetaQUAST (parameters: --min-contig 1000 -k-mer-stats -extensive-mis-size 100) based on the alignment of assembled contigs against reference genomes are used to evaluate the accuracy of misassembly identification (See Additional file 2: Table S12-13 for more details of these datasets).

Four real datasets from GAGE-B project [30], including *Bacillus cereus ATCC 10987* (*B. cereus*), *Mycobacterium abscessus 6G-0125-R* (*M. abscessus*), *Rhodobacter sphaeroides 2.4.1* (*R. sphaeroides*), and *Vibrio cholerae CP1032* (*V. cholerae*), are used to see the performance of metaMIC on isolate genomes. All the trimmed Illumina paired-end reads can be downloaded from http://ccb.jhu.edu/gage_b/genomeAssemblies/index.html. The contigs used in our experiments are assembled by Velvet [31], and all the Velvet assembled contigs can be downloaded from http://bioinf.spbau.ru/en/content/spades-30-gage-b-data-sets. These contigs are further scaffolded with BESST [33] and ScaffMatch [34]. The accuracy of both contigs and scaffolds are evaluated with QUAST [9] as gold standard. The reads are mapped to contigs in all datasets with the help of BWA-MEM (v.0.7.17).

### Comparison to competing methods

We compare metaMIC against two popular approaches, i.e., ALE and DeepMAsED, for identifying misassembled contigs in metagenomic datasets. ALE calculates likelihood scores for each individual position in the contig based on the likelihood of observed reads generated from a given assembly, and the score consists of four sub-scores (placement, insert, depth, and *k*-mer). When applying ALE to identify misassembled contigs, we aggregated the four ALE sub-scores into a contig score similar to Kuhring et al. [14]. In more detail, we select a threshold for each ALE sub-score. For each contig, a position whose sub-score smaller than the threshold is counted as a potential error, then these error counts were summed up and normalized by the contig length and the total number of sub-scores as the final score. The thresholds of four ALE sub-scores were chosen that can lead to its best performance on the training dataset. Specifically, we computed the AUPRC for each metagenome in the training dataset given the input thresholds. We then computed the average AUPRC scores of all metagenomes for each combination of

the input thresholds. We selected the thresholds leading to the highest AUPRC score on the training dataset, and the thresholds considered in ALE can be found in Table S8. DeepMAsED is a recently developed method for identifying misassembled contigs in metagenomic assemblies. We re-trained the DeepMAsED model on the training dataset used by metaMIC with recommended parameters, and the re-trained model was used to predict misassemblies in the benchmark datasets (See Additional file 2: Table S11 for the performance of default model provided by DeepMAsED on the benchmark datasets). Given the strong class imbalance problem for each dataset that contains limited number of misassemblies, we used AUPRC to evaluate the performance of these methods. We performed a more thorough comparison with other tools, and compared metaMIC against VALET [15], FRCbam [44], NucBreak [45], and Pilon [46] to identify misassembled contigs. We evaluated these tools with the same BAM files as those used for metaMIC, and all tools were run with their default settings. Note that all these tools described above have never been evaluated on metagenomics datasets. As shown in Table S9, metaMIC outperforms all the other tools significantly with the highest *F*1 score.

When recognizing misassembly breakpoints in metagenomic contigs, we only compare metaMIC with ALE since DeepMAsED cannot localize the misassembly breakpoints. For ALE, the four sub-scores per base are summed up and the position with a lower ALE score is a likely assembly error. Here, a position with the lowest ALE score across a given misassembled contigs is regarded as a misassembly breakpoint for ALE. Similar to metaMIC, we do not consider the 300-bp region at the both ends of the contigs for ALE.

### Evaluation of binning results

When evaluating a set of bins reconstructed from simulated microbial datasets, we use BLASTn to map each bin against the ground truth genomes used for each dataset. A representative genome of each bin is determined based on the genome which can be covered by the highest fraction of nucleotides from that bin. Then for each bin, we define the number of nucleotides in the bin that belong to the representative genome as true positives (TP). The total number of nucleotides from the bin not covered by the representative genome corresponds to the false positives (FP), and the number of nucleotides in the representative genome not covered by any contigs from that bin represents the false negatives (FN). Then the completeness, contamination, and *F*1 score of each bin can be calculated as follows.

$$completeness = \frac{TP}{TP + FN}$$

$$purity = \frac{TP}{TP + FP}$$

$$contamination = 1 - purity$$

$$F1\ score = \frac{2 * completeness * purity}{completeness + purity}$$

For the real metagenomics data sets where the ground truth genomes are inaccessible, we employ CheckM [47] to estimate the completeness and contamination of each bin.

### Validating misassemblies identified by metaMIC with long-read assemblies

A combined rumen fluid and solid sample sequenced with both short- and long-read DNA sequencing technologies (SRA BioProject numbers *PRJNA507739*) are used to validate the metaMIC predictions on short-read assemblies. Short-read sequencing approaches are based on Illumina NextSeq 500 sequencing technology, and long-read sequencing is performed on the Sequel systems via PacBio's single molecule real-time (SMRT) sequencing technology. In total, the sample contains approximately 143 million short reads and 600,000 long reads. The Illumina short reads are assembled into contigs with MEGAHIT while the long-read assemblies are generated with PacBio reads by using metaFlye [48]. Then the short-read assemblies were evaluated through using long-read assemblies as gold standard with MetaQUAST.

### Abbreviations
MAG        Metagenome-assembled genomes
bp         Base pair
AUPRC      Area under the precision-recall curve
KAD        *k*-mer abundance difference
AUC        Area under the receiver operating characteristic curve

## Supplementary Information

> Additional file 1: Fig. S1-S29.
>
> Additional file 2: Table S1-S13.
>
> Additional file 3. Review history.

**Review history**
The review history is available as Additional file 3.

**Peer review information**
Andrew Cosgrove was the primary editors of this article and managed its editorial process and peer review in collaboration with the rest of the editorial team.

**Authors' contributions**
X.Z conceived the study and supervised the project. S.L, X.Z, W.C, and L.P.C designed the method. S.L and S.P wrote the software. S.L and C.S designed and performed the analysis. S.L wrote the first draft of the manuscript. All authors contributed to the revision of manuscript prior to submission and all authors read and approved the final manuscript.

**Funding**
This work was partly supported by the National Key Research and Development Program of China (No. 2020YFA0712403 to X.M.Z, No. 2019YFA0905601 to W.H.C), National Natural Science Foundation of China (NSFC) (Nos. T2225015, 61932008), and Shanghai Municipal Science and Technology Major Project (No. 2018SHZDZX01, 2021YFF0703703).

**Availability of data and materials**
All the data used for validation is publicly available and can be downloaded from NCBI databases. Raw sequencing data of three real metagenomic datasets can be obtained from the Sequence Read Archive (SRA) under BioProject accession number *PRJNA485056* [25], *PRJNA504891* [26], and *PRJNA507739* [48], while the raw sequencing data from the mock metagenomic community (GIS20) can be obtained from the SRA under BioProject accession number *PRJEB29139* [29]. The source code describing our methods is freely available under the MIT license at GitHub (https://github.com/ZhaoXM-Lab/metaMIC) [49] and Zenodo (https://doi.org/10.5281/zenodo.7263041) [50].

## Declarations

### Ethics approval and consent to participate
Not applicable.

### Consent for publication
Not applicable.

### Competing interests
The authors declare that they have no competing interests.

### Author details
[1]Institute of Science and Technology for Brain-Inspired Intelligence, Fudan University, Shanghai, China. [2]Key Laboratory of Molecular Biophysics of the Ministry of Education, Hubei Key Laboratory of Bioinformatics and Molecular-imaging, Center for Artificial Intelligence Biology, Department of Bioinformatics and Systems Biology, College of Life Science and Technology, Huazhong University of Science and Technology, Wuhan, Hubei, China. [3]MOE Key Laboratory of Computational Neuroscience and Brain-Inspired Intelligence, and MOE Frontiers Center for Brain Science, Fudan University, Shanghai, China. [4]College of Life Science, Henan Normal University, Xinxiang, Henan, China. [5]State Key Laboratory of Medical Neurobiology, Institutes of Brain Science, Fudan University, Shanghai, China. [6]Research Institute of Intelligent Complex Systems, Fudan University, Shanghai, China. [7]International Human Phenome Institutes (Shanghai), Shanghai, China. [8]Zhangjiang Fudan International Innovation Center, Shanghai, China.

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

## 