## [Additional file 3. Review history. · Genome Biology]

Review History

First round of review

Reviewer 1

Were you able to assess all statistics in the manuscript, including the appropriateness of statistical tests used? No

Were you able to directly test the methods? Yes

Comments to author:

The problem the authors want to address (i.e. identification of misassembled contigs and fine-grained localization of breakpoints in these) is definitely of interest and important in order to improve metagenomic sequence analyses, but their approach appears to suffer from important technical weaknesses that -in my opinion- limit its current usefulness:

- i) The model building and evaluation process is not properly described, e.g. how many cross-validations were carried out and what are the associated performance metrics? The authors do not describe (not even mention) n-fold cross-validation experiments, characterize learning errors, precise the choice of hyperparameters,...
- ii) Random Forests can be sensitive to strong class imbalances in the datasets as here, and it is unclear that downsampling alone did handle these biases.
- iii) The authors only compare the performance of the first tier (detection of misassembled contigs) of their approach against two alternative methods without describing how the latter were used/parameterized, e.g. which DeepMAsED model was used or how was it re-trained and tested (the AUPR plots shown might indicate these tools were not properly used)? A more thorough comparison could also include some of the very large number of tools available, e.g. misassembly detection by analyzing reads remapping against assembly include (among others) AMOSValidate, REAPR, FRCbam, Pilon, VALET, while methods computing the probability of the reads given the assembly include CGAL and LAP beyond ALE (for which the authors "manually choose a threshold for each sub-score").
- iv) The second tier of the approach (localization of misassembly breakpoints) could be improved beyond the criteria of max of breakpoint ratios used by the authors, as there is a vast literature and research already available on change point detection algorithms.
- v) It is unclear the differences presented in Figures 4A and 5A (application of the method on real datasets using bin quality measures) are statistically significant.
- vi) It would be interesting to know the relative importance of the various (and overlapping) features extracted from the alignment for model training. In this respect, there appears to be some confusing statements, e.g. line 337 indicates four types of features are mainly used but line 347 suggests six read features are used for training classifier.
- vii) Overall, there appears to be a lack of formal description of the approach, e.g. there is no quantitative information on the training process and selecting 5kb+ contigs from metagenomic assemblies might indicate very low amount of data is ultimately used; as contig-based features are used, further details on the effect of contig length, normalization procedures and the handling of multi-error containing contigs would be welcomed; how exactly were multi-assigned reads handled; several sentences would warrant deeper

clarifications, e.g. "The read coverage and fragment coverage are further standardized as the ones divided by the means of the corresponding coverages of all bases across the contig or a given region",...

viii) I may be wrong, but the term "intra-species" should mean "strain-level" imho, but the authors use the term in the context of isolate genome assemblies.

Therefore I feel more formal descriptions of the method are required, along with more careful and extensive experiments and comparisons, in order to support the conclusion of the authors and a broad usage of their tool.

Reviewer 2

Were you able to assess all statistics in the manuscript, including the appropriateness of statistical tests used? Yes: The statistical analysis in the manuscript is appropriate and sufficient.

Were you able to directly test the methods? Yes

Comments to author:

The paper "metaMIC: reference-free Misassembly Identification and Correction of de novo metagenomic assemblies" by Lai et al. describes a method for the reference-free identification and correction of misassembled metagenomic contigs. The paper described a solution to very relevant and somewhat neglected problem in microbial genomics, the detection of misassembled, often chimeric, contigs using a machine learning based method that can additionally can split the misassembled contigs accurately at their breakpoints. Benchmarking on simulated datasets, isolate genomes and real world dataset including one with long reads as groundtruth shows that the new method outperforms existing methods . Moreover the authors show that the accurate splitting of misassemblies can help to improve the binning of metagenomics assembled genomes. Overall the paper is very clear and well written, the description of the methods, the analysis and results are sound. The paper is important for people in the metagenomic field and will improve the quality of future metagenomic studies.

Nevertheless are a few points that the authors should comment on:

- The authors define two new metrics to define error regions, the anomaly score and read breakpoint ratio that can be used to define the breakpoints of misassemblies. Error regions are defined by an anomaly score > 0.95 . Have the authors tested different score thresholds and could they discuss the impact ? The same holds for the 100bp sliding window. Can the window size be changed ?
- It would be interesting to see, where the breakpoints are located, within coding regions, intergenic or/and in repeat regions ?
- Overall it would be important for the field if the authors could also provide a model for the metaSpades, which is (also according CAMI) with the Megahit currently the most widely used assembler whereas idba-ud is not used that often anymore in recent metagenomic studies
- The method assumes a normal distribution of the insert sizes of the mate pairs. Is the expected insert size estimated from the bam file or can it also be given by the user ?

- For the random forest model: from the reading no cross validation was used, why ? The authors should report on the performance of the RF classifier: OoB error, importance values, ..

- In general it would also be good to test non-simulated dataset for training and/or testing like the SIHUMI (Simplified HUMAN Interstitial Microbiota (SIHUMI), <https://microbiomejournal.biomedcentral.com/articles/10.1186/s40168-021-01035-8#ref-CR49>)

- One feature the authors is contig coverage that is known to be also highly affected from replication state of the bacteria. Could the authors comment on this ?

- Are the detected misassemblies all from bacterial genomes (in the real datasets)? What about archaea, eukaryotes or bacteria not in training data/with unusual genome features etc? Can you comment on this ?

- 435ff: For all CAMI datasets a minimum threshold for contigs you use 5000bp, why so large ?

Minor comments

- For the GAGE-B dataset the authors use Velvet for the assembly, but they do only provide models

- As metaMIC tool heavily relies on bwa and samtools for information extraction and preprocessing it would be nice to package it either in a workflow tool like Snakemake

- Line 227ff: how many metaMIC predictions you could validate with the PacBio long reads (percentage of all corrections ? How many did you falsely correct if any at all ? Could you discuss this a bit more ?

- line 405ff: can you change these default values (splitting score and minimal contig length) in the software ? Are the break points are still reported even if they are too small, this would be very helpful

- line 419ff: filtering for CheckM completeness of only 50% might still contain quite some low quality mags, any other checks to ensure high quality of input data?

- line 424ff: can the training be done while keeping the short contigs ? ( apparently yes, with -l flag, correct ?)

- line 310: many bacterium typo

- Could the authors add information about runtimes ?

Installing/running the metaMIC tool as described on <https://github.com/ZhaoXM-Lab/metaMIC>:

* scipy, scikit-learn, requests dependencies missing (as conda)

* the 'readthedocs' link not working

* needed to get bioconda openssl=1.0 for samtools mpileup to run

* had to manually load megahit model

* it would be nice to get the corrected assembly as a whole, not just a fasta file with corrected contigs, which you would have to manually integrate into the assembly while removing the split/missassembled ones. or maybe a small utility function to generate it on demand in metaMIC

Reviewer #1:

***Comment 1:** The model building and evaluation process is not properly described, e.g. how many cross-validations were carried out and what are the associated performance metrics? The authors do not describe (not even mention) n-fold cross-validation experiments, characterize learning errors, precise the choice of hyperparameters,...*

Response: Thank you for your suggestion. We are sorry for not describing the model training procedure clearly. We performed 10-fold cross-validations on the training set to select the optimal hyperparameters for the random forest classifier. In more detail, given 30 metagenomes as the training set, we randomly pick 3 of them for validation while the rest 27 metagenomes for training. This procedure will be repeated for 100 times to make sure the robustness of the model trained.

As shown in Figure R1, to handle the imbalance problem existing between the positive and negative samples, we will perform down-sampling to obtain the same number of correct contigs as that of the misassembled contigs, where there are fewer misassembled contigs as positive samples. Specifically, a subset of negative samples will be randomly selected from the whole negative training set during the down-sampling procedure and the size of the negative subset is the same as the positive training set. This down-sampling will be repeated for 10 times, and a classifier will be trained based on the positive training set and a selected negative subset each time. As shown in Figure R1 below, an ensemble classifier will be obtained with the average result of the 10 trained classifiers as output, and the ensemble classifier will be used as the final classifier.

We used the area under the precision-recall curve (AUPRC) to evaluate the performance of the trained classifier, and the parameters leading to the highest AUPRC during the cross-validation procedure will be used as the optimal hyperparameters (see Table S6). In addition to AUPRC, area under the receiver operating characteristic curve (AUC) and out-of-bag (Oob) error rate were also used to evaluate metaMIC. The results of the 10-fold cross-validation experiments were provided in Table S7 and the precision-recall curves were shown in Figure S23, where the 10-fold cross-validation results averaged over 100 times were shown. As shown in Table S7, the average AUPRC scores for the IDBA_UD and MEGAHIT assembled training datasets were 0.71 and 0.55, respectively.

The procedure we described here for handling sample imbalance, training model with cross-validation and optimizing parameters has been well known and widely used tricks in machine learning [1]. We have added “Model training and optimization”

section on Pages 20-21 in the revised manuscript for detailed description of the model building and evaluation procedure.

Figure R1. The schematic flowchart of training model on the down-sampled negative samples for identifying mis-assembled contigs, where each individual classifier will be trained with 10-fold cross-validations and the optimization of the hyperparameters will be performed in the outer loop of the cross-validation procedure.

Comment 2: *Random Forests can be sensitive to strong class imbalances in the datasets as here, and it is unclear that down-sampling alone did handle these biases.*

Response: As mentioned above, we adopt down-sampling to handle the imbalance problem (Figure R1). To ensure the robustness of the resulting classifier, the down-sampling procedure will be repeated for 10 times, and the resulting ensemble classifier will be obtained. To look at the robustness of the classifier, we show below the performance of the ensemble classifier trained over different number of negative subsets down-sampled (Figure R2). It can be seen that the performance of the ensemble classifier tends to be stable after the down-sampling procedure is repeated for more than

10 times. Therefore, the down-sampling will be performed 10 times, and an ensemble classifier will be obtained based on the 10 individual random forest classifiers trained on the 10 negative subsets.

Figure R2. The performance of the ensemble classifier trained on different number of down-sampled negative subsets for (a) MEGAHIT-specific training dataset and (b) IDBA_UD-specific training dataset.

Comment 3: *The authors only compare the performance of the first tier (detection of misassembled contigs) of their approach against two alternative methods without describing how the latter were used/parameterized, e.g. which DeepMAsED model was used or how was it re-trained and tested (the AUPR plots shown might indicate these tools were not properly used)?*

Response: We thank the reviewer for the comment. As for DeepMAsED, the results reported in our manuscript were obtained with the default model trained in DeepMAsED. For fair comparison, we have tried to re-train and optimize the DeepMAsED model with our training datasets used by metaMIC based on the training module provided by DeepMAsED with recommended parameters. Unfortunately, DeepMAsED does not work on our training set, and the training process fail to converge. As shown in the emails we communicated with the author of DeepMAsED (Figure R3), the author response once but not response to the failure of DeepMAsED to work on our training set. Therefore, we still use the results by the default model of DeepMAsED for comparison with metaMIC. Even though the training set used by DeepMAsED is not exactly the same as that used by metaMIC, we argue that the comparison is still fair since the training set used to train metaMIC was generated in the same way as that used by DeepMAsED [2], where the 30 simulated bacterial metagenomes were generated with the same pipeline and the same 1,000 reference genomes selected from the Genome Taxonomy Database as DeepMAsED did.

As for ALE, we used it for identifying misassembled contigs by following Kuhring et al [3]. The output of ALE is a score averaged over four contig-length normalized sub-scores, i.e. *depth*, *place*, *insert* and *kMer*, and the score is the likelihood of a contig misassembled.

The detailed descriptions of how these tools were used were provided in “Comparison to competing methods” section on Pages 25-26.

Figure R3. the emails we communicated with the author of DeepMAseD.

Comment 4: A more thorough could also include some of the very large number of tools available, e.g. misassembly detection by analyzing reads remapping against assembly include (among others) AMOSValidate, REAPR, FRCbam, Pilon, VALET, while methods computing the probability of the reads given the assembly include CGAL and LAP beyond ALE (for which the authors "manually choose a threshold for each sub-score").

Response: In light of your comment, we performed a more thorough comparison with other tools, and compared metaMIC against NucBreak [4], FRCbam [5], Pilon [6] and VALET [7] to identify misassembled contigs. We excluded REAPR [8] and AMOSValidate [9] from comparison as they are no longer maintained. A recently developed tool called NucBreak [4] has been demonstrated to perform better than REAPR, and FRCbam [5] is a tool developed based on AMOSValidate that uses the similar misassembly features to track assembly errors across assembled contigs. Therefore, it is not necessary to compare with REAPR and AMOSValidate after we compare metaMIC against NucBreak and FRCbam. We also excluded CGAL [10] and LAP [11] from consideration since these two tools are used for evaluating assembling quality instead of detecting assembling errors, where both CGAL and LAP can be used to evaluate the overall quality of assembled genomes but fail to evaluate the quality of each contig. We evaluated these tools with the same BAM files as those used for metaMIC, and all tools were run with their default settings. Note that all these tools described above have never been reported to be evaluated on metagenomics datasets. As shown in the Table R1 below, metaMIC outperforms all the other tools significantly with the highest *F1* score. The benchmark results of these tools were described in the “Comparison to competing methods” section of Pages 25-26 in the revised manuscript.

Table R1. Comparison of metaMIC against NucBreak, FRCbam, Pilon and VALET when identifying misassembled contigs on CAMI datasets. Pilon was not evaluated on the CAMI-High dataset due to the long running time and large memory requirement.

Dataset	Tool	Recall	Precision	F1
CAMI-Medium	NucBreak	0.17	0.50	0.25
	FRCbam	0.40	0.20	0.26
	Pilon	0.03	0.20	0.06
	VALET	0.03	0.40	0.05
	metaMIC	0.51	0.55	0.53
CAMI-High	NucBreak	0.85	0.08	0.15
	FRCbam	0.34	0.10	0.16
	Pilon	\	\	\
	VALET	0.04	0.28	0.07
	metaMIC	0.70	0.36	0.47
CAMI-Skin	NucBreak	0.17	0.34	0.23
	FRCbam	0.60	0.23	0.33

	Pilon	0.03	0.35	0.06
	VALET	0.06	0.30	0.11
	metaMIC	0.77	0.35	0.48
CAMI-Oral	NucBreak	0.18	0.50	0.25
	FRCbam	0.56	0.20	0.30
	Pilon	0.09	0.24	0.13
	VALET	0.03	0.21	0.05
	metaMIC	0.77	0.35	0.48
CAMI-Gut	NucBreak	0.17	0.38	0.24
	FRCbam	0.57	0.23	0.41
	Pilon	0.03	0.23	0.05
	VALET	0.03	0.31	0.06
	metaMIC	0.80	0.34	0.48

***Comment 5:** The second tier of the approach (localization of misassembly breakpoints) could be improved beyond the criteria of max of breakpoint ratios used by the authors, as there is a vast literature and research already available on change point detection algorithms.*

Response: We thank the reviewer for the suggestion. Indeed, there are some change point detection (CPD) algorithms available in literature, and we have tried to use them for the localization of misassembly breakpoints. Here, we have tried several popular CPD algorithms, including Pruned Exact linear Time (PELT) search method [10], Binary segmentation (Binseg) search method [11], Window-based (Window) search method and Dynamic programming (Dynp) search method (using the Python ruptures package [12]).

Here, all the CPD algorithms were applied with the same features as those used by metaMIC, i.e. the window-based features including four different types: read coverage, nucleotide variants, k -mer abundance difference and read pair consistency. As shown in Figure R4 below, although the CPD methods can work for the localization of misassembly breakpoints, metaMIC outperforms them significantly with much smaller error sizes.

Figure R4. The distribution of error size (left) and normalized error size (right) of the misassembly breakpoints recognized by different approaches on the CAMI-High dataset, where the cost function leading to the highest accuracy for each CPD method is chosen here. The three cost functions *l1*, *l2* and *rbf* denotes least absolute deviation, least squared deviation and kernelized mean change, respectively.

Comment 6: *It is unclear the differences presented in Figures 4A and 5A (application of the method on real datasets using bin quality measures) are statistically significant.*

Response: Thank you for your comment. In addition to comparing the number of obtained high-quality bins (Completeness >90 and Contamination <5) before and after correction, we also performed paired Wilcoxon’s test to statistically compare the bin-wise *F1* scores of corrected bins with those of corresponding original bins (see Table R2 below). The bin-wise *F1* score was calculated based on the completeness and purity of each bin, where the completeness was equal to recall while the purity corresponded to the precision. From the results, we can see that both the number of high-quality bins and the bin-wise *F1* scores improve significantly after metaMIC correction, implying that metaMIC correction can really help improve binning results. In our revised manuscript, we have added these results in paragraph 1 of Page 10 and paragraph 1 of Page 11.

Table R2. Comparison of MetaBAT2 binning results before and after metaMIC correction on both simulated and real datasets.

		Mean of bin-wise F1		Paired Wilcoxon’s test
Dataset		Original	Corrected	P -value
Simulated datasets	Sim-Virome	0.64	0.75	4.69e-3
	CAMI-Medium	0.73	0.76	2.93e-4

	CAMI-Skin	0.52	0.53	3.53e-2
	CAMI-Oral	0.29	0.30	0.51
	CAMI-Gut	0.58	0.60	0.07
Human gut metagenomes	Ethiopian	0.75	0.79	2.37e-3
	Madagascar	0.80	0.82	3.07e-5

Comment 7: It would be interesting to know the relative importance of the various (and overlapping) features extracted from the alignment for model training.

Response: Thank you for your suggestion. In light of your comment, we have calculated the relative importance of features as shown in Figure S24. We found that the top important features mainly belong to the read pair consistency based features for both IDBA_UD and MEGAHIT assembled contigs although the top ranked features are slightly different for each assembler specific model. The proportion of discordant reads with their mates mapped to different contigs (discordant_loc_ratio) was ranked first in MEGAHIT specific model, while the proportion of proper reads (proper_read_ratio) was the top first important feature in IDBA_UD specific model. The difference between top features of assembler specific models may be explained by the different distribution of misassembly types in the contigs generated by different assemblers. For instance, inter-genome translocation generally causes discordant read pairs with different mapped locations, and it was also indeed the most common assembly error generated by MEGAHIT (see Figure S6). To validate our guess, we have looked at the top features for metaSPAdes-specific model and noticed that the top features are also different from those of MEGAHIT-specific and IDBA_UD-specific models. We have discussed this in Additional file 1 (see “Relative importance of input features” section on Page 6).

Fig. S24 Ranking of feature importance of the random forest classifier trained on (a) MEGAHIT-specific training dataset, (b) IDBA_UD-specific training dataset and (c) metaSPAdes-specific training dataset, respectively.

Comment 8: *In this respect, there appears to be some confusing statements, e.g. line 337 indicates four types of features are mainly used but line 347 suggests six read features are used for training classifier.*

Response: We are sorry for the confusion. All features used by metaMIC can be categorized into four feature types (as stated in *line 337*), including read coverage, read pair consistency, nucleotide variants and *k*-mer abundance consistency. The six read features in *line 347* refer to the features belonging to the feature type of read pair consistency, including `proper_read_ratio`, `supplementary_read_ratio`, `inversion_read_ratio`, `discordant_loc_ratio`, `inversion_read_ratio` and `discordant_size_ratio`. We have revised the text to make it more clear and easier to understand in the revised manuscript (“Features extracted from reads and contigs” section on Page 18) and Additional file 1 (see “Contig-based features” section on Pages 2-6).

Comment 9: *Overall, there appears to be a lack of formal description of the approach, e.g. there is no quantitative information on the training process and selecting 5kb+ contigs from metagenomic assemblies might indicate very low amount of data is ultimately used; as contig-based features are used, further details on the effect of contig length, normalization procedures and the handling of multi-error containing contigs would be welcomed; how exactly were multi-assigned reads handled; several sentences would warrant deeper clarifications, e.g. "The read coverage and fragment coverage are further standardized as the ones divided by the means of the corresponding coverages of all bases across the contig or a given region",...*

Response: Thank you for your suggestions. In the revised manuscript, more detailed descriptions of the approach have been provided as partly shown below, including (1) training process, (2) selection of 5kb+ contigs, (3) normalization procedure, (4) the handling of multi-error containing contigs, (5) the handling of multi-assigned reads, and so on.

(1) The detailed description of training process has been mentioned above (see response to comment 1), and was provided in our revised manuscript (see ‘Model training and optimization’ section on Pages 20-21).

(2) We chose 5kb+ contigs from metagenomic assemblies based on the following reasons: i) The goal of metaMIC is to identify misassemblies, such as inversion, relocation and translocation in the contigs, instead of small assembly errors (local

misassembly). We have investigated the misassemblies occurring in the CAMI dataset, and noticed that approximately 70~80% of misassemblies happen in 5kb+ contigs (see Figure S27) and the probability of a contig misassembled increases as the contig becomes longer (see Figure S28); ii) Based on the above observation, metaMIC was trained and tested on contigs of length 5kb+. In light of your comment, we also tested metaMIC on contigs shorter than 5kb, where metaMIC still performs best among all tools (see Figure S30 and “Evaluating metaMIC on contigs shorter than 5,000bp” section in Additional file 1). Although we trained metaMIC with contigs longer than 5kb, metaMIC allows users to train and test their models on shorter contigs or contigs of whatever length. This issue was also discussed in the revised manuscript (paragraph 2 on Page 24).

(3) For the features used by metaMIC, we have standardized them by taking into account the contig length. For instance, the features of read coverage and fragment coverage at each position i of contig c are standardized as follows similarly to [13].

$$\text{Standardized_read_coverage}_{ic} = \frac{\text{read_coverage}_{ic}}{(\sum_{j=1}^{L_c} \text{read_coverage}_{jc})/L_c},$$

$$\text{Standardized_fragment_coverage}_{ic} = \frac{\text{fragment_coverage}_{ic}}{(\sum_{j=1}^{L_c} \text{fragment_coverage}_{jc})/L_c},$$

where L_c is the length of contig c . Then the standardized deviations of coverage and fragment coverage for contig c are used as input of metaMIC as shown below.

$$\sigma_{\text{read_coverage}_c} = \sqrt{\frac{\sum_{j=1}^{L_c} (\text{standardized_read_coverage}_{jc} - 1)^2}{L_c}},$$

$$\sigma_{\text{fragment_coverage}_c} = \sqrt{\frac{\sum_{j=1}^{L_c} (\text{standardized_fragment_coverage}_{jc} - 1)^2}{L_c}}.$$

(4) Right now, metaMIC only works on misassembled contigs containing a single misassembly error since multi-error containing contigs only make up a small proportion (~5%) of all misassembled contigs (see Figure S22). Another reason that metaMIC does not work on multi-error containing contigs is that the correction of misassembled contigs by metaMIC mainly relies on splitting at misassembly breakpoints. If there are multiple breakpoints, the correction may fail to work and lead to many short and fragmented sequences. In the future, we will extend metaMIC to work on multi-error containing contigs, where the correction will be accomplished by joining the broken contigs into new contigs instead of only splitting misassembled contigs into fragments. This issue is also discussed in our revised manuscript (paragraph 3 on Page 15).

(5) samtools is generally used to filter low quality mappings and ensure each read aligned perfectly. Here, the same strategy has been used so that no reads can map equally well to more than one location in the metagenomic assembly.

Comment 10: *I may be wrong, but the term "intra-species" should mean "strain-level" imho, but the authors use the term in the context of isolate genome assemblies.*

Response: We thank the reviewer for pointing out this error. We have carefully corrected “intra-species misassemblies” as “intra-genome misassemblies” in the context of isolate genome assemblies (paragraph 2 of Page 12, lines 254-255).

Reviewer #2:

Comment 1: *The authors define two new metrics to define error regions, the anomaly score and read breakpoint ratio that can be used to define the breakpoints of misassemblies. Error regions are defined by an anomaly score >0.95 . Have the authors tested different score thresholds and could they discuss the impact? The same holds for the 100bp sliding window. Can the window size be changed?*

Response: Thanks for your comment. The threshold of anomaly score >0.95 was reported as it leads to the highest $F1$ score (see Figure S25). As shown in Figure S25, a lower anomaly score can identify more misassemblies, however, more false positive predictions will be introduced. In contrast, a higher threshold of anomaly score reduces the number of false positives but leads to lower recall. In fact, a user can set the threshold of anomaly score with the parameter `--at` in metaMIC as he/she wish. We recommend users to set the threshold of anomaly score within the range of 0.9 to 0.95.

We used the sliding window of 100bp for localizing misassembly breakpoints as reported in literature [8]. In light of your comment, we have tested the reliability of metaMIC with different sliding window sizes ranging from 100bp to 500bp on the CAMI datasets. As shown in Figure S21 of Additional file 1, the accuracy of localizing misassembly breakpoints significantly decreased as the sliding window size increases, thus a shorter window size had a higher resolution to pinpoint error regions. However, more computation resources will be required if the sliding window becomes shorter. Therefore, in metaMIC, the size of sliding window is fixed to 100bp. This issue has been discussed in our revised manuscript (paragraph 3 of Page 15, lines 311-314).

Comment 2: *It would be interesting to see, where the breakpoints are located, within coding regions, intergenic or/and in repeat regions?*

Response: Thanks for your suggestion. In light of your comment, we have calculated the proportion of breakpoints located in coding sequences (CDS), intergenic and repeat regions, respectively. The CDS and intergenic regions were predicted by Prodigal (v2.6.3) [14], whereas the repeat regions were determined by TRF (v4.07) [15] and RepeatScout (v1.0.6) [16]. As shown in the Figure R5 below, approximately 65% of breakpoints in the CAMI datasets were located in CDS regions, indicating that most misassemblies could disrupt gene structures. Note that the vast majority of lineages in prokaryotes have over 80% CDS density [17]. Repetitive sequences, which can occur in both noncoding and coding region of genomes, usually affect the continuity and correctness of assemblies [18]. We found that more than 10% of breakpoints were located within repeat regions, and ~35% of misassembled contigs contained short tandem repeats (STRs), which far exceeds the correctly assembled contigs containing STRs (Fisher test, $p\text{-value} < 2.2e-16$, in CAMI-Medium dataset). Therefore, the repetitive characteristic in contigs can be regarded as a candidate misassembly signature, and will be taken into account in our future work. We also discussed this in our revised manuscript (paragraph 3 of Page 7, paragraph 2 of Page 16).

Figure R5. The proportion of misassembly breakpoints located in coding sequences (CDS), intergenic and repeat regions in the CAMI datasets.

Comment 3: *Overall it would be important for the field if the authors could also provide a model for the metaSpades, which is (also according CAMI) with the Megahit currently the most widely used assembler whereas idba-ud is not used that often anymore in recent metagenomic studies*

Response: We thank the reviewer for this insightful suggestion. We have added a model for contigs assembled by metaSPAdes in the metaMIC, and the users can use the

parameter *-a metaSPAdes* to identify misassemblies in contigs assembled by metaSPAdes. The results of the 10-fold cross-validation experiments on metaSPAdes-assembled training dataset were provided in Table S7 and the precision-recall curves were shown in Figure S23, where the 10-fold cross-validation results averaged over 100 times were shown. We also evaluated the performance of metaMIC on the CAMI datasets where contigs were assembled by metaSPAdes, and the associated performance was shown in Figure S7 (paragraph 2 of Page 7 in revised manuscript).

Fig. S7 Precision-recall curves for identifying misassembled contigs in the CAMI datasets, where contigs from CAMI datasets were assembled by metaSPAdes. CAMI-High was excluded due to insufficient memory for metaSPAdes assembling.

Comment 4: *The method assumes a normal distribution of the insert sizes of the mate pairs. Is the expected insert size estimated from the bam file or can it also be given by the user?*

Response: The expected insert size is estimated from the bam file directly and is not provided by the user. Only read pairs satisfying the following conditions are considered for insert size estimation: 1) Both reads are mapped to the same contig; 2) The reads have different orientation relative to the contig. For each paired-end read $r(r_l, r_r)$, the insert size is calculated as follows.

$$Insert_size = P_{re} - P_{ls} + 1, \text{ where}$$

P_{ls} : the position of left mate read r_l starting at the contig

P_{re} : the position of right mate read r_r ending at the contig

Then the expected insert size (μ) was calculated as the median value of all insert sizes, whereas the standard deviation (σ) of insert sizes was estimated by the median absolute deviation of insert sizes similarly to [13]. The detailed description of insert size estimation was provided in Additional file 1 (see “Insert size estimation” section of Page 1).

Comment 5: *For the random forest model: from the reading no cross validation was used, why? The authors should report on the performance of the RF classifier: OoB error, importance values, ..*

Response: Thank you for your comment. We performed 10-fold cross-validations on the training set to select the optimal hyperparameters for the random forest classifier. In more detail, given 30 metagenomes as the training set, we randomly pick 3 of them for validation while the rest 27 metagenomes for training. This procedure will be repeated for 100 times to make sure the robustness of the model trained.

The details of cross-validation and imbalance problem please refer to our response to Reviewer 1’s Comment 1, and the importance of different features please refer to our response to Reviewer 1’s Comment 7. The performance of different tools was evaluated with the area under the precision-recall curve (AUPRC), area under the receiver operating characteristic curve (AUC) and out-of-bag (Oob) error rate as shown in Table S7.

Table S7. The 10-fold cross-validation results for each assembler-specific training dataset, where the training contigs are assembled by IDBA_UD, MEGAHIT and metaSPAdes, respectively.

Assembler	AUPRC	AUC	Oob error rate
IDBA_UD	0.71	0.93	0.12
MEGAHIT	0.55	0.90	0.16
metaSPAdes	0.54	0.87	0.18

Comment 6: *In general it would also be good to test non-simulated dataset for training and/or testing like the SIHUMI (Simplified HUMAN Interstinal Microbiota (SIHUMI)), <https://microbiomejournal.biomedcentral.com/articles/10.1186/s40168-021-01035-8#ref-CR49>*

Response: Thank you very much for your suggestion. The metagenomes in the SIHUMI are only from the fecal samples instead of the synthetic dataset sequenced from known microbes (see Figure R6) and the assembling errors are not known, and cannot be used for testing. Thus, we evaluated metaMIC on another mock microbiome community (GIS20) which were sequenced from a mixture of 20 cultured strains of bacteria (NCBI project ID PRJEB29139) [19]. The performance of metaMIC on the GIS20 dataset were shown in Figure S18, where metaMIC significantly outperforms existing tools in identifying misassembled contigs and localizing misassembly breakpoints. Approximately 65% misassembly breakpoints can be localized with predicted error size smaller than 500bp (Manuscript: paragraph 1 of Page 12, lines 245-251; Additional file 1: see “Application to mock community” section of Pages 8-9).

Figure R6. The emails we communicated with the author of SIHUMI.

Fig. S18 The performance of metaMIC for identifying misassemblies in the GIS20 datasets, where contigs were assembled by MEGAHIT and IDBA_UD, separately. The first two rows show the benchmark results of metaMIC against ALE and DeepMAseD for identifying misassembled contigs. The last row shows the distribution of error size (left) and normalized error size (right) of misassembly breakpoints recognized by metaMIC and ALE.

Comment 7: One feature the authors is contig coverage that is known to be also highly affected from replication state of the bacteria. Could the authors comment on this?

Response: Thank you for your comment. Indeed, the coverage of different regions that are far apart in a bacterial genome may exhibit significant difference if the microbe is growing rapidly. Given the gradual decline of coverage from the origin to terminus of replication of a bacterium [20], the difference of sequencing coverage for adjacent genomic regions is relatively small. Therefore, if two regions from different locations

of a bacterial genome are wrongly assembled into one contig, it will exhibit differential coverage patterns, whereas the read coverage should be consistent over the entire contig if it is perfectly assembled. The same works for regions from different bacterial genomes but are wrongly assembled into one contig. Taken together, the change-point in the contig coverage feature may indicate a possible relocation or translocation. So the contig coverage feature will not be affected by replication state of the bacteria theoretically.

Comment 8: *Are the detected misassemblies all from bacterial genomes (in the real datasets)? What about archaea, eukaryotes or bacteria not in training data/unusual genome features etc? Can you comment on this?*

Response: Thanks for your comment. As metagenomes are primarily composed of bacteria and approximately 98% of the classified reads are assigned to bacteria [21], the misassemblies detected by metaMIC were mainly from bacterial genomes in the real datasets. However, metaMIC can be easily transferred to archaea or eukaryotes since it only relies on the features extracted from the mapping results between paired-end reads and assembled contigs. For example, we have shown that the metaMIC model trained on the bacterial metagenomes can work well on simulated virome datasets (Fig. 2d). In the future, we will consider training metaMIC specifically for archaea or eukaryotes. As for bacteria not in the training set, we have evaluated metaMIC across multiple independent test sets and it performs very well. We also discussed this in our revised manuscript (paragraph 1 of Page 15, lines 297-301).

Comment 9: *435ff: For all CAMI datasets a minimum threshold for contigs you use 5000bp, why so large?*

Response: Thanks for your comment. We chose the 5,000bp as the minimum threshold of contigs mainly based on the following reasons.

(i) The goal of metaMIC is to identify misassemblies, such as inversion, relocation and translocation in the contigs, instead of small assembly errors (local misassembly). We have investigated the misassemblies occurring in the CAMI dataset, and noticed that approximately 70~80% of misassemblies happen in 5kb+ contigs (see Figure S27) and the probability of a contig misassembled increases as the contig becomes longer (see Figure S28);

(ii) Based on the above observation, metaMIC was trained and tested on contigs of length 5kb+. In light of your comment, we also tested metaMIC on contigs shorter than 5kb, where metaMIC still performs best among all tools (see Figure S30 and

“Evaluating metaMIC on contigs shorter than 5,000bp” section in Additional file 1). Although we trained metaMIC with contigs longer than 5kb, metaMIC allows users to train and test their models on shorter contigs or contigs of whatever length. This issue was also discussed in the revised manuscript (paragraph 1 on Page 24, lines 490-492).

Minor comments

Comment 10: For the GAGE-B dataset the authors use Velvet for the assembly, but they do only provide models

Response: We added more detailed descriptions on the dataset in the revised manuscript (paragraph 2 on Page 24, lines 499-502). The trimmed paired-end reads, assembled contigs and reference genomes are all available in the GAGE-B project. All the trimmed Illumina paired-end reads can be downloaded from http://ccb.jhu.edu/gage_b/datasets/index.html, and all the Velvet assembled contigs can be downloaded directly from <http://bioinf.spbau.ru/en/content/spades-30-gage-b-data-sets>.

Comment 11: As metaMIC tool heavily relies on bwa and samtools for information extraction and preprocessing it would be nice to package it either in a workflow tool like Snakemake

Response: We thank the reviewer for this insightful suggestion. In the current version, metaMIC can be easily installed from source, and both samtools and bwa are available in Bioconda. We will consider to package them together with metaMIC in future.

Comment 12: Line 227ff: how many metaMIC predictions you could validate with the PacBio long reads (percentage of all corrections? How many did you falsely correct if any at all? Could you discuss this a bit more?)

Response: Thanks for your comment. When using the long-read assemblies as gold standards, 10% of short-read assemblies can be aligned to long-read assemblies, among which a total of 93 misassembled contigs were detected. When working on all short-read assemblies, metaMIC is able to identify 537 misassemblies, where 43 misassemblies can be validated by long-read assemblies. In other words, these 43 misassemblies belong to the 93 misassembled contigs. Additionally, approximately 80% of all misassembly breakpoints in these 43 misassembled contigs can be accurately predicted. None of the contigs that were correctly assembled validated by long-read assemblies were wrongly identified as misassemblies by metaMIC. Therefore, for

metaMIC, the precision is 100% and the recall is 43/93=46%. We also discussed this in our revised manuscript (paragraph 2 of Page 11, lines 231-234).

***Comment 13:** line 405ff: can you change these default values (splitting score and minimal contig length) in the software? Are the break points are still reported even if they are too small, this would be very helpful*

Response: Yes, the user can use the parameter `-s` (Minimum length of split fragments) to change the minimal splitting length, whereas the splitting score can be adjusted by the parameter `--st` (Threshold of contig score for correcting misassemblies in metagenomes). metaMIC will report all misassembly breakpoints regardless of where they were localized, and the breakpoints can still be reported even if the split fragments are too short. However, the breakpoints located within the 300bp region at both ends of a contig will not be reported as 300bp regions at both ends of a contig are not considered in the metaMIC due to poor mapping quality. As for splitting score, metaMIC only reports misassembly breakpoints for contigs with scores larger than the selected threshold.

***Comment 14:** line 419ff: filtering for CheckM completeness of only 50% might still contain quite some low quality mags, any other checks to ensure high quality of input data?*

Response: Thank you for your comments. The distributions of completeness and contamination of selected reference genomes used for generating training datasets are shown in Figure S26, and the completeness of the selected reference genomes was actually no less than 90% while the CheckM contamination were all below 5%. Thus, the quality of selected reference bacterial genomes could be guaranteed. In the revised manuscript, we have modified the “CheckM estimated completeness >50” to “CheckM estimated completeness >90” (paragraph 2 of Page 23, line 474).

Fig. S26 CheckM-estimated completeness and contamination of 1,000 bacterial genomes used for generating training datasets.

Comment 15: line 424ff: can the training be done while keeping the short contigs? (-> apparently yes, with *-l* flag, correct?)

Response: Yes, the user can use the parameter *-l* (Minimal contig length) to keep the short contigs when performing training process.

Comment 16: line 310: many bacterium typo

Response: We thank the reviewer for pointing this out and corrected “many bacterium” to “many bacteria” (paragraph 2 of Page 16, lines 335-336).

Comment 17: Could the authors add information about runtimes?

Response: Thank you for your suggestion. In light of your comment, we have evaluated the running time and memory usage of metaMIC on both single genomic and metagenomic assemblies (see Table S5). In the updated manuscript, we have added the information about runtimes and memory usage (paragraph 2 on Page 17, lines 352-353).

Table S5. Running time and memory usage of metaMIC on single genomic/metagenomic assemblies from GAGE-B datasets, CAMI2-datasets and metagenomes from *Ethiopian* and *Madagascar* cohort. For the metagenomes from real datasets, we randomly chose 10 samples to evaluate the running time and memory usage of metaMIC. metaMIC was run on CPU machine (AMD EPYC 7601 32-Core processor, 756G memory in server).

GAGE-B		CAMI2			Real metagenomes	
B.cereus	M.abscessus	Skin	Oral	Gut	Ethiopian	Madagascar

	Read size (G)	1	0.74	30	30	30	8	6
	BAM size (G)	0.27	0.22	13	13	13	1.7	1.1
Feature extraction	Time (min)	9	5	660	646	694	360	370
	Memory (MB)	629	630	6,033	10,209	6,805	900	645
Prediction	Time (min)	0.73	1	3	5	2	1	1
	Memory (MB)	160	155	3,048	3,298	3,748	2,904	2,900

References

1. Zhao XM, Li X, Chen L, Aihara K: **Protein classification with imbalanced data.** *Proteins* 2008, **70**:1125-1132.
2. Mineeva O, Rojas-Carulla M, Ley RE, Scholkopf B, Youngblut ND: **DeepMAseD: evaluating the quality of metagenomic assemblies.** *Bioinformatics* 2020, **36**:3011-3017.
3. Kuhring M, Dabrowski PW, Piro VC, Nitsche A, Renard BY: **SuRankCo: supervised ranking of contigs in de novo assemblies.** *Bmc Bioinformatics* 2015, **16**.
4. Khelik K, Sandve GK, Nederbragt AJ, Rognes T: **NucBreak: location of structural errors in a genome assembly by using paired-end Illumina reads.** *BMC Bioinformatics* 2020, **21**:66.
5. Vezzi F, Narzisi G, Mishra B: **Reevaluating assembly evaluations with feature response curves: GAGE and assemblathon.** *PLoS One* 2012, **7**:e52210.
6. Walker BJ, Abeel T, Shea T, Priest M, Abouelliel A, Sakthikumar S, Cuomo CA, Zeng Q, Wortman J, Young SK, Earl AM: **Pilon: an integrated tool for comprehensive microbial variant detection and genome assembly improvement.** *PLoS One* 2014, **9**:e112963.
7. Olson ND, Treangen TJ, Hill CM, Cepeda-Espinoza V, Ghurye J, Koren S, Pop M: **Metagenomic assembly through the lens of validation: recent advances in assessing and improving the quality of genomes assembled from metagenomes.** *Brief Bioinform* 2019, **20**:1140-1150.
8. Hunt M, Kikuchi T, Sanders M, Newbold C, Berriman M, Otto TD: **REAPR: a universal tool for genome assembly evaluation.** *Genome Biol* 2013, **14**:R47.
9. Phillippy AM, Schatz MC, Pop M: **Genome assembly forensics: finding the elusive mis-assembly.** *Genome Biol* 2008, **9**:R55.
10. Dorcas Wambui G: **The Power of the Pruned Exact Linear Time(PELT) Test in Multiple Change-point Detection.** *American Journal of Theoretical and Applied Statistics* 2015, **4**.

11. Scott AJ, & Knott, M.: **A cluster analysis method for grouping means in the analysis of variance.** *Biometrics* , 30, 507-512 1974.
12. Truong C, Oudre L, Vayatis N: **Selective review of offline change point detection methods.** *Signal Processing* 2020, 167.
13. Wu B, Li M, Liao X, Luo J, Wu F, Pan Y, Wang J: **MEC: Misassembly Error Correction in contigs based on distribution of paired-end reads and statistics of GC-contents.** *IEEE/ACM Trans Comput Biol Bioinform* 2018.
14. Hyatt D, Chen GL, Locascio PF, Land ML, Larimer FW, Hauser LJ: **Prodigal: prokaryotic gene recognition and translation initiation site identification.** *BMC Bioinformatics* 2010, 11:119.
15. Benson G: **Tandem repeats finder: a program to analyze DNA sequences.** *Nucleic Acids Res* 1999, 27:573-580.
16. Price AL, Jones NC, Pevzner PA: **De novo identification of repeat families in large genomes.** *Bioinformatics* 2005, 21 Suppl 1:i351-358.
17. Kuo CH, Ochman H: **Deletional bias across the three domains of life.** *Genome Biol Evol* 2009, 1:145-152.
18. Torresen OK, Star B, Mier P, Andrade-Navarro MA, Bateman A, Jarnot P, Gruca A, Grynberg M, Kajava AV, Promponas VJ, et al: **Tandem repeats lead to sequence assembly errors and impose multi-level challenges for genome and protein databases.** *Nucleic Acids Res* 2019, 47:10994-11006.
19. Bertrand D, Shaw J, Kalathiyappan M, Ng AHQ, Kumar MS, Li C, Dvornicic M, Soldo JP, Koh JY, Tong C, et al: **Hybrid metagenomic assembly enables high-resolution analysis of resistance determinants and mobile elements in human microbiomes.** *Nat Biotechnol* 2019, 37:937-944.
20. Korem T, Zeevi D, Suez J, Weinberger A, Avnit-Sagi T, Pompan-Lotan M, Matot E, Jona G, Harmelin A, Cohen N, et al: **Growth dynamics of gut microbiota in health and disease inferred from single metagenomic samples.** *Science* 2015, 349:1101-1106.
21. Song Y, Himmel B, Ohrmalm L, Gyarmati P: **The Microbiota in Hematologic Malignancies.** *Curr Treat Options Oncol* 2020, 21:2.

Second round of review

Reviewer 1

While I appreciate the authors' efforts to clarify some of the items raised previously, I feel unsatisfied by their answers to two of my main concerns:

i) In order to clarify the model building and evaluation process, the authors mention that 10-fold cross validation was used during the training of individual random forest classifier, but it still remains unclear how the model was actually evaluated. In k-fold cross-validation, the training set is split into k different training and validation sets, and the performance across partitions is compared to select the best hyperparameters, which is what the authors have apparently done. However and importantly, a test (i.e. hold-out) set (e.g. 10% of the data) should be used to assess the performance of the learned model on data not used for training or validation; this test set should be used at the end of the process to avoid fitting the (ensemble) model to the test set.

Unfortunately, this prevents a sound evaluation of the model's generalization ability; a comparison of the error rate on the training dataset versus the validation dataset could further have been informative with respect to the overfitting issue, which could be relevant given the authors mention that the performance of the models appears assembler-dependent.

ii) The comparison of metaMIC against other tools still looks problematic to me (e.g. figure 2); i understand that the authors encountered difficulties when training the DeepMAsED model on the metaMIC datasets, and that this led them to measure the performance of DeepMAsED trained on its original dataset, but this does not ultimately guarantee a fair comparison. The comparison against the ALE tool also remains disputable as this tool was not designed for this kind of contig-level analysis, which required some manual intervention by the authors.

To me, the main trouble with these comparisons is that they lead to very strong differences (see figure 2) between the performance of metaMIC versus the other tools. In my experience, such behaviour frequently reveals a problem with the testing process itself.

Reviewer 2

The authors gave detailed answers to all open questions, discussed the comments and made appropriate changes to the manuscript.

It would be great if the authors could update their github page to the newest version, e.g. the link to the new models webpage https://zenodo.org/record/5768805#.YdbprCwo_sc and the docs page is somehow not available.

Reviewer #1:

***Comment 1:** In order to clarify the model building and evaluation process, the authors mention that 10-fold cross validation was used during the training of individual random forest classifier, but it still remains unclear how the model was actually evaluated. In k -fold cross-validation, the training set is split into k different training and validation sets, and the performance across partitions is compared to select the best hyperparameters, which is what the authors have apparently done. However and importantly, a test (i.e. hold-out) set (e.g. 10% of the data) should be used to assess the performance of the learned model on data not used for training or validation; this test set should be used at the end of the process to avoid fitting the (ensemble) model to the test set.*

Unfortunately, this prevents a sound evaluation of the model's generalization ability; a comparison of the error rate on the training dataset versus the validation dataset could further have been informative with respect to the overfitting issue, which could be relevant given the authors mention that the performance of the models appears assembler-dependent.

Response: Thank you for your suggestion and sorry for not making it clear. In the revised manuscript, we present the evaluation results of our metaMIC on both validation and test sets. We split the dataset into two parts: 2/3 of the data as training set for optimizing the parameters with the cross-validation, and 1/3 of the data as independent test set for assessing the performance of the learned models. As shown in Figure R1, we performed 10-fold cross-validations on the training set to optimize the parameters and the procedure has been described very clearly in our previous response letter, and this procedure will be repeated for 100 times to make sure the robustness of the model trained, where down-sampling was performed for the negative training set to handle the imbalance problem between positive and negative training set. Then we evaluated the performance of the learned ensemble classifier on the independent test set that was not used for training. Both the results of the 10-fold cross-validation experiments on the training set and the results on the test set were provided in Table R1, where our metaMIC has shown very robust performance. The detailed description of both training and evaluation procedure was provided in “Model training and optimization” section on Pages 21-22.

Figure R1. The schematic flowchart of training and testing metaMIC, where the dataset was split into training and test set. Both the parameter optimization and down-sampling of negative samples were performed on training set, and the resulting model will be evaluated on the independent test set.

Table R1. The cross-validation and test results of metaMIC on each assembler-specific dataset, where the training contigs are assembled by IDBA_UD, MEGAHIT, and metaSPAdes, respectively. The test set was not seen during the training procedure (10-fold cross-validations).

	Assembler	AUPRC	AUC
Training set (10-fold cross-validation)	IDBA_UD	0.72	0.93
	MEGAHIT	0.54	0.90
	metaSPAdes	0.54	0.87
Test set	IDBA_UD	0.62	0.91
	MEGAHIT	0.54	0.89
	metaSPAdes	0.53	0.86

AUPRC: area under the precision-recall curve; AUC: area under the receiver operating characteristic curve;

***Comment 2:** The comparison of metaMIC against other tools still looks problematic to me (e.g. figure 2); i understand that the authors encountered difficulties when training the DeepMAsED model on the metaMIC datasets, and that this led them to measure the performance of DeepMAsED trained on its original dataset, but this does not ultimately guarantee a fair comparison. The comparison against the ALE tool also remains disputable as this tool was not designed for this kind of contig-level analysis, which required some manual intervention by the authors.*

To me, the main trouble with these comparisons is that they lead to very strong differences (see figure 2) between the performance of metaMIC versus the other tools. In my experience, such behaviour frequently reveals a problem with the testing process itself.

Response: Thanks for your comment. In light of your comment, for fair comparison, we trained DeepMAsED on the training dataset used to train metaMIC and optimized the parameters of DeepMAsED by following its original paper, and we also optimized the parameters of ALE on the same training set. We can see the performance of DeepMAsED and ALE indeed improves after the optimization of their parameters.

As for DeepMAsED, we re-trained the DeepMAsED model on our training datasets used by metaMIC. As the train module provided by DeepMAsED does not work on our training set, we re-implemented the DeepMAsED model with the same network architecture with PyTorch (The original DeepMAsED model was achieved by Keras) by following its original paper [1]. We have tried to optimize the parameters for DeepMAsED but noticed that the model with its recommended parameters performs best, and we therefore used the default parameters here and each model was trained for 20 epochs.

As for ALE, instead of manually choosing the thresholds for ALE sub-scores (i.e. *depth*, *place*, *insert* and *kMer*) as we did previously in the original manuscript, we choose the thresholds that led to the best performance of ALE on the training dataset used by metaMIC. In more detail, for each metagenome in the training dataset (30 metagenomes), we computed the area under the precision-recall curve (AUPRC) given the input thresholds. We then computed the average AUPRC scores on all 30 metagenomes for each combination of thresholds. We selected the thresholds leading to the highest AUPRC score on the training dataset, and the thresholds considered in ALE can be found in Table S8.

As shown in Table R2, both the results of DeepMAsED and ALE were improved significantly after parameter optimization or model re-training with our training datasets (highlighted in yellow). We noticed that DeepMAsED always performs better than ALE over all benchmark datasets, which was consistent with the conclusion from the DeepMAsED paper [1]. metaMIC outperforms all the other tools significantly with the highest AUPRC score even when ALE and DeepMAsED were re-trained or optimized. We have updated the results of DeepMAsED and ALE in Figure 2, and the detailed description of how these tools were used were provided in “Comparison to competing methods” section on Pages 25-26.

Table R2. The AUPRC scores of DeepMAsED, ALE and metaMIC in identifying misassembled contigs over the benchmark datasets. We show the results of DeepMAsED and ALE before and after re-training or optimizing with our training datasets, respectively.

Dataset	DeepMAsED		ALE		metaMIC
	before	after	before	after	
CAMI-Medium	0.08	0.28	0.10	0.22	0.55
CAMI-High	0.12	0.26	0.13	0.21	0.48
CAMI-Skin	0.05	0.37	0.14	0.23	0.53
CAMI-Oral	0.05	0.38	0.14	0.23	0.46
CAMI-Gut	0.03	0.34	0.11	0.34	0.51
Sim-Virome	0.13	0.29	0.17	0.28	0.58

Reviewer #2:

Comment 1: It would be great if the authors could update their github page to the newest version, e.g. the link to the new models webpage https://zenodo.org/record/5768805#.YdbprCwo_sc and the docs page is somehow not available.

Response: Thank you for your comment. We have updated the github page in the <https://github.com/ZhaoXM-Lab/metaMIC>.

References

1. Mineeva O, Rojas-Carulla M, Ley RE, Scholkopf B, Youngblut ND: **DeepMAsED: evaluating the quality of metagenomic assemblies.** *Bioinformatics* 2020, **36**:3011-3017.

Third round of Review

Reviewer 1

Though I appreciate the efforts of the authors to address previous remarks, these don't dissipate most of my previous concerns. On the methodological level, I was surprised in the first round of review that there was no mention of cross-validation (cv) in the model construction and evaluation process, the absence of hold-out dataset in their later answer etc.

The authors mention a 10-fold cv so I'm still very puzzled by the sentence "This procedure (cv) will be repeated for 100 times to make sure the robustness of the model trained", while the average of the performance in the cv loop or something like that would probably be more expected.

On the other hand, I'm wondering why the authors did not provide the absolute number (only proportions are reported) of 5KB+ contigs used in the training despite being asked, or more detailed information (e.g. the real empirical values) on the insert sizes used in the experiments, which have probably a significant impact on the results.

Beyond the methodological issues, I'm feeling that their manuscript falls a little bit short for Genome Biology because similar methods (including those used in the comparisons) were originally published in journals with more moderate impact (with the exception of two methods not considered by the authors, e.g REAPR was published in GB and is available from [1] even if it is probably no longer actively maintained as indicated by the authors).

[1] <https://www.sanger.ac.uk/tool/reapr/>

I believe the work of the authors is certainly valuable and of practical value, and hope this feedback will be useful to improve this manuscript.

Reviewer #1:

***Comment 1:** On the methodological level, I was surprised in the first round of review that there was no mention of cross-validation (cv) in the model construction and evaluation process, the absence of hold-out dataset in their later answer etc.*

Response: We are sorry for not mentioning the cross-validation procedure in the initial submission, since the cross-validation (CV) is a commonly used trick in machine learning for choosing the optimal parameters or evaluating the classifier trained. We originally focused more on the evaluations of our tool metaMIC and ignored to mention the training details in the paper. Thanks to your suggestion in your first-round review, we gave the details on how the CV performed, and which parameters to be chosen in the supplementary table S6. Furthermore, we also gave the details on how to handle the imbalance problem when training the classifier.

***Comment 2:** The authors mention a 10-fold cv so I'm still very puzzled by the sentence "This procedure (cv) will be repeated for 100 times to make sure the robustness of the model trained", while the average of the performance in the cv loop or something like that would probably be more expected.*

Response: When using the 10-fold CV, the samples will be randomly split into 10 subsets without overlap, which means that the 10-subsets when performing the 10-fold CV this time will be different from the 10-subsets obtained from the 10-fold CV next time. Therefore, in machine learning, the 10-fold CV will be performed many times to make the results obtained are robust. We have given the average of the performance in the CV loop in Supplementary Table S7 (see Additional file 1) in our previous version. We have also mentioned this and gave detailed results when we responded to your comments previously.

***Comment 3:** On the other hand, I'm wondering why the authors did not provide the absolute number (only proportions are reported) of 5Kb+ contigs used in the training despite being asked, or more detailed information (e.g. the real empirical values) on the insert sizes used in the experiments, which have probably a significant impact on the results.*

Response: Thanks for your suggestion. The absolute number of 5Kb+ contigs as well as the number of misassembled contigs for each dataset used in our experiments were

provided in the Table R1, and all the information can also be found in Supplementary Table S12 (see Additional file 1).

In addition, for each dataset, we also provided the detailed information on the real insert size as well as the expected insert size (μ) that we estimated directly from the bam file in Table R2, which can also be found in Supplementary Table S13 (see Additional file 1). As shown in Table R2, the estimated insert size is very close to the true insert size.

Firstly, as metaMIC only relies on the expected insert size (μ) and standard deviation (σ) of all insert sizes (distance between two mates mapping to the contigs) that were estimated directly from the mapping BAM file instead of the real insert size to identify discordant reads with wrong insert size, the real insert size actually have no significant impact on the results. Secondly, our model trained on the training set with 200 bp (The default parameter in the MGSIM that we used for simulating metagenomics) insert size could still generalize well to the CAMI datasets with 270 bp insert size.

Table R1. The absolute number of >5Kb+ contigs as well as the number of misassembled contigs for each assembler-specific dataset used in the experiments.

	Assembler	No. of total contigs (>5Kb+)	No. of misassembled contigs
Training set	MEGAHIT	114,248	9,755
	IDBA_UD	11,3239	8,928
	metaSPAdes	46,334	3,216
Test set	MEGAHIT	58,174	5,039
	IDBA_UD	59,738	4,903
	metaSPAdes	23,793	1,633
CAMI-Medium	MEGAHIT	15,842	1,369
	IDBA_UD	13,961	928
	metaSPAdes	12,955	384
CAMI-High	MEGAHIT	27,532	2,285
	IDBA_UD	11,942	1,017
	metaSPAdes	\	\
CAMI-Skin	MEGAHIT	12,015	723
	IDBA_UD	5,900	668
	metaSPAdes	10,480	133
CAMI-Oral	MEGAHIT	10,026	840
	IDBA_UD	6,374	445
	metaSPAdes	6,190	171
CAMI-Gut	MEGAHIT	11,863	838

	IDBA_UD	7,104	483
	metaSPAdes	8,845	149

Table R2. The detailed information for datasets used in the experiments including read length, real insert size (mean), expected insert size (μ) as well as the estimated standard deviation (σ).

	Read length (bp)	Real insert size (bp)	Assembler	Expected insert size (μ)	Standard deviation (σ)
Training set	150	200	MEGAHIT	199.0	10.38
			IDBA_UD	233.0	31.13
			metaSPAdes	240.0	22.23
Test set	150	200	MEGAHIT	199.0	10.38
			IDBA_UD	234.0	31.13
			metaSPAdes	240.0	22.23
CAMI-Medium	150	270	MEGAHIT	269.0	26.69
			IDBA_UD	269.0	26.69
			metaSPAdes	269.0	26.69
CAMI-High	150	270	MEGAHIT	267.0	28.17
			IDBA_UD	264.0	31.13
			metaSPAdes	\	\
CAMI-Skin	150	270	MEGAHIT	269.0	26.69
			IDBA_UD	269.0	28.17
			metaSPAdes	264.0	31.13
CAMI-Oral	150	270	MEGAHIT	269.0	26.69
			IDBA_UD	268.0	28.17
			metaSPAdes	264.0	31.13
CAMI-Gut	150	270	MEGAHIT	269.0	26.69
			IDBA_UD	269.0	28.17
			metaSPAdes	264.0	31.13

The expected insert size (μ) was calculated as the median value of all insert sizes, whereas the standard deviation (σ) was estimated by the median absolute deviation of insert sizes

Comment 4: Beyond the methodological issues, I'm feeling that their manuscript falls a little bit short for Genome Biology because similar methods (including those used in the comparisons) were originally published in journals with more moderate impact (with the exception of two methods not considered by the authors, e.g REAPR was published in GB and is available from [1] even if it is probably no longer actively maintained as indicated by the authors).

Response: Firstly, we would like to argue that there are few methodologies that have been developed for error detection in metagenomic assembly. Those you have mentioned previously and asked us to compare, are actually either not for metagenomic assembly or for genome assembly. Even so, we have compared our metaMIC against them as you required to the good performance of metaMIC. Secondly, REAPR is developed to identify assembly errors in the context of isolate genomes. Given the uneven coverage (different abundance) of different microbial species in the metagenomic context, REAPR doesn't work for metagenomic assembly. In fact, VALET [1] adapts the approaches within REPAR and extends it to the context of metagenomics by binning contigs before applying REPAR, as shown in the flowchart of the VALET pipeline in Figure R1 below. We have already compared metaMIC against VALET in our previous manuscript, where metaMIC always outperforms VALET with significantly higher *FI* scores in all datasets as shown in the Supplementary Table S9 (see Additional file 1).

Fig. R1. Overview of the VALET pipeline.

1. Olson ND, Treangen TJ, Hill CM, Cepeda-Espinoza V, Ghurye J, Koren S, Pop M: **Metagenomic assembly through the lens of validation: recent advances in assessing and improving the quality of genomes assembled from metagenomes.** *Briefings in Bioinformatics* 2019, **20**:1140-1150.